# Benchmarking and optimization of methods for the detection of identity-by-descent in high-recombining *Plasmodium falciparum* genomes

**Bing Guo[1,2], Shannon Takala-Harrison[1]\*[†], Timothy D O'Connor[2]\*[†]**

[1]Center for Vaccine Development and Global Health, University of Maryland School of Medicine, Baltimore, United States; [2]Institute for Genome Sciences, University of Maryland School of Medicine, Baltimore, United States

*For correspondence:
stakala@som.umaryland.edu
(ST-H);
timothydoconnor@gmail.com
(CDO)

[†]These authors contributed
equally to this work

Reviewing Editor: Bavesh
D Kana, University of the
Witwatersrand, South Africa

## eLife Assessment

This **important** study presents an evaluation of several tools used for detecting Identity-By-Descent (IBD) segments in highly recombining genomes, using simulated data to replicate the high recombination and low marker density of *Plasmodium falciparum*, the parasite responsible for malaria. The evidence presented by the authors is **convincing** demonstrating that users should be cautious calling IBD when SNP density is low and recombination rate is high. This study will be of interest to scientists working in the field of genome evolution and infectious diseases

**Abstract** Genomic surveillance is crucial for identifying at-risk populations for targeted malaria control and elimination. Identity-by-descent (IBD) is increasingly being used in *Plasmodium* population genomics to estimate genetic relatedness, effective population size ($N_e$), population structure, and signals of positive selection. Despite its potential, a thorough evaluation of IBD segment detection tools for species with high recombination rates, such as *Plasmodium falciparum*, remains absent. Here, we perform comprehensive benchmarking of IBD callers – probabilistic (`hmmIBD`, `isoRelate`), identity-by-state-based (`hap-IBD`, `phased IBD`) and others (`Refined IBD`) – using population genetic simulations tailored for high recombination, and IBD quality metrics at both the IBD segment level and the IBD-based downstream inference level. Our results demonstrate that low marker density per genetic unit, related to high recombination relative to mutation, significantly compromises the accuracy of detected IBD segments. In genomes with high recombination rates resembling *P. falciparum*, most IBD callers exhibit high false negative rates for shorter IBD segments, which can be partially mitigated through optimization of IBD caller parameters, especially those related to marker density. Notably, IBD detected with optimized parameters allows for more accurate capture of selection signals and population structure; IBD-based $N_e$ inference is very sensitive to IBD detection errors, with IBD called from `hmmIBD` uniquely providing less biased estimates of $N_e$ in this context. Validation with empirical data from the MalariaGEN *Pf7* database, representing different transmission settings, corroborates these findings. We conclude that context-specific evaluation and parameter optimization are essential for accurate IBD detection in high-recombining species and recommend `hmmIBD` for *Plasmodium* species, especially for quality-sensitive analyses, such as estimation of $N_e$. Our optimization and high-level benchmarking methods not only improve IBD segment detection in high-recombining genomes but also enhance overall genomic analysis, paving the way for more accurate genomic surveillance and targeted intervention strategies for malaria.

## Introduction

Malaria is a mosquito-borne disease caused by *Plasmodium* parasites. It poses a significant public health challenge globally, with an estimated 263 million clinical cases and 597,000 deaths occurring in 2023 (*World Health Organization, 2024*). Despite intensive efforts toward malaria control and elimination, malaria reduction has slowed or plateaued in recent years, due to multiple factors including antimalarial drug resistance and lack of highly efficacious vaccines and detailed, timely surveillance. Advances in sequencing technologies and the scale of resequencing studies now allow for parasite genomic surveillance, which can provide insights into the efficacy of malaria interventions and guide the design of targeted elimination strategies in different transmission settings (*Neafsey et al., 2021*; *Wesolowski et al., 2018*; *Shetty et al., 2019*).

Identity-by-descent (IBD) is an essential tool in population genomics that has been used to estimate genetic relatedness (*Taylor et al., 2019b*; *Gerlovina et al., 2022*; *Henden et al., 2018*; *Schaffner et al., 2023*), positive selection (*Henden et al., 2018*; *Guo et al., 2024*; *Browning and Browning, 2020*), effective population size ($N_e$) (*Gutenkunst et al., 2009*; *Morgan et al., 2020*), fine-scale population structure (*Guo et al., 2024*; *Shetty et al., 2019*; *Nait Saada et al., 2020*), and migration patterns (*Shetty et al., 2019*; *Al-Asadi et al., 2019*). IBD-based analyses of parasite genomic data have applications in diverse epidemiological settings, enhancing malaria surveillance and control by providing crucial information to researchers and policymakers (*Wesolowski et al., 2018*; *Camponovo et al., 2023*; *Guo et al., 2025*). In high transmission settings, a rapid decrease in genetic diversity and effective population size may indicate a successful malaria intervention (*Morgan et al., 2020*). In intermediate transmission settings, IBD-based analysis of parasite population structure and migration provides valuable insights into sources and sinks of transmission for planning targeted elimination strategies (*Shetty et al., 2019*; *Guo et al., 2024*; *Henden et al., 2018*). In low transmission settings, IBD-based estimates of pedigree relationship analysis between infections can help differentiate local transmission from importation and causes of recurrent infection (*Taylor et al., 2019b*; *Wong et al., 2025*). Additionally, IBD-based detection of positive selection can assist in identifying and monitoring the emergence and spread of antimalarial drug resistance (*Henden et al., 2018*; *Amambua-Ngwa et al., 2019*; *Guo et al., 2024*).

However, the reliability of IBD-based analysis is highly dependent on the accuracy of the detected IBD segments. Insufficient density of genetic markers, on a local or genome-wide scale, probably contributes to high error rates in the identification of IBD segments (*Browning and Browning, 2020*; *Freyman et al., 2021*; *Zhou et al., 2020*) and reduced accuracy of IBD-based estimates of population demography (*Browning and Browning, 2015*; *Taylor et al., 2019a*). Many IBD detection methods have been designed for human genomes, where the demographic history and evolutionary parameters, including the recombination rate, differ considerably from *Plasmodium falciparum* (*Pf*). The effective population size of humans has increased rapidly in recent history (*Gutenkunst et al., 2009*), while that of *Pf* is decreasing, particularly in regions such as Southeast Asia and South America (*Joy et al., 2003*; *World Health Organization, 2024*), due to enhanced malaria elimination efforts. More importantly, *Pf* genomes recombine about 70 times more frequently per unit of physical distance (*Gardner et al., 2002*; *Su et al., 1999*; *Jiang et al., 2011*; *Amambua-Ngwa et al., 2019*; *Miles et al., 2016*) than the human genome (*Kong et al., 2002*), while sharing a similar mutation rate (*Bopp et al., 2013*; *Camponovo et al., 2023*; *Churcher et al., 2014*; *Hamilton et al., 2017*; *Huber et al., 2016*; *McDew-White et al., 2019*; *Neafsey et al., 2021*) as human genomes (*Campbell et al., 2012*) on the order of $10^{-8}$ per base pair per generation. The decreasing population size (*Guo et al., 2024*) and the high recombination rate in *Pf* result in a reduced number of common variants, such as single-nucleotide polymorphisms (SNPs), per unit of genetic distance. Large human whole-genome sequencing data sets typically provide millions of common biallelic SNP variants (*Taliun et al., 2021*), while *Pf* data sets only have tens of thousands (*Abdel Hamid et al., 2023*). Given that the human genome is about twice as large as *Pf* in genetic units, the per-centimorgan (cM) SNP density in *Pf* can be two orders of magnitude lower than in humans, which may not provide sufficient information for detecting IBD segments. Thus, it is critical to understand whether IBD detection methods can still generate accurate IBD segments under low SNP density conditions, considering the specific evolutionary parameters of the *Pf* genome.

Evaluating the quality of the detected IBD segments requires benchmarking with the known ground truth through simulation studies (*Zhou et al., 2020*; *Freyman et al., 2021*; *Tang et al., 2022*;

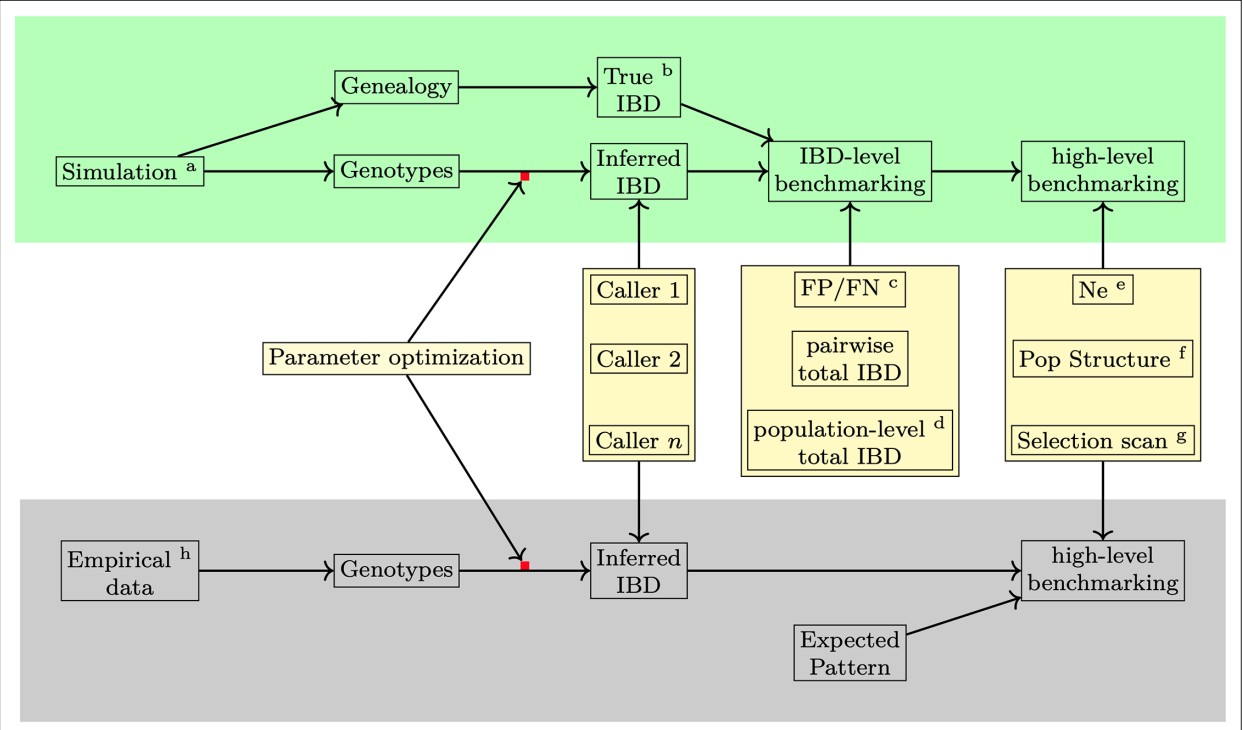

**Figure 1.** Overview of methods used in benchmarking IBD detection methods. Benchmarking and optimization of IBD callers for *Pf* include simulation analyses (top, green shading) and empirical data-based validation analyses (bottom, gray shading). (1) For the simulation study, the genealogical trees and phased genotype data are generated via the combination of forward (SLiM *Haller et al., 2019*; *Haller and Messer, 2019*) and coalescent (msprime *Baumdicker et al., 2022*) simulations (indicated by the superscript **a** in the diagram). True IBD is obtained from a simulated genealogy tree via tskibd (*Guo et al., 2024*) (**b**) and inferred IBD from the phased genotype using different IBD callers, including hap-IBD (*Zhou et al., 2020*), hmmIBD (*Schaffner et al., 2018*), isoRelate (*Henden et al., 2018*), Refined IBD (*Browning and Browning, 2013*), and phased IBD (*Freyman et al., 2021*). IBD benchmarking is performed at two levels. The first is at the IBD segment level. The metrics include false positive and false negative rates (**c**), population-level total IBD per length bin (*Browning and Browning, 2015*) (**d**), and total IBD per isolate pair. The second is at the level of IBD-based downstream analyses, including the effective population size ($N_e$) by IBDNe (*Browning and Browning, 2015*) (**e**), community membership through the InfoMap algorithm (*Rosvall et al., 2009*; *Csardi and Nepusz, 2006*) (**f**), and selection signals by statistics $X_{iHS}$ (*Guo et al., 2024*) (**g**). As the default parameters of different IBD callers, particularly those developed for human data, may not be ideal for *Pf* genomes, we performed grid searches for key parameters for each IBD caller so that the comparison was based on the best performance of each caller (see *Supplementary file 1—Data S1* for detailed information on simulation and IBD calling parameters and used values). (2) For validation in empirical data where true IBD is not available, we obtained IBD-based estimates using IBD from different callers and assessed which version of IBD can generate the expected patterns. The empirical data sets are subsampled from the MalariaGEN *Pf7* database (*Abdel Hamid et al., 2023*) (**h**).

*Shemirani et al., 2021*). The performance of IBD detection tools developed for use in the human context is typically measured using simulated genomes reflecting demographic and evolutionary parameters of human genomes (*Browning and Browning, 2011*; *Browning and Browning, 2011*; *Zhou et al., 2020*; *Tang et al., 2022*; *Freyman et al., 2021*; *Shemirani et al., 2021*), which likely do not apply directly to *Pf*. For tools explicitly designed for malaria parasites, such as isoRelate and hmmIBD, the evaluation of the quality of IBD was based on parent-offspring (*Schaffner et al., 2018*) or pedigree-based simulations (up to 25 generations; *Henden et al., 2018*) that focused primarily on close relatives, which more likely mirrors low malaria transmission settings than high transmission settings. Furthermore, benchmarking methods and definitions of IBD accuracy are inconsistent across studies (*Zhou et al., 2020*; *Freyman et al., 2021*; *Schaffner et al., 2018*; *Henden et al., 2018*), making the results of the quality evaluation of IBD difficult to compare. Considering the limitations of existing evaluations of IBD detection methods for *Pf* genomes, a unified benchmarking framework specifically designed for high recombining *Pf* genomes from low- and high-transmission settings is needed. Such a framework will assist researchers in comparing and prioritizing different IBD detection methods for intended downstream analysis.

In the present study, we developed a unified IBD benchmarking framework that reflects the demographic and evolutionary parameters of *Pf* (*Figure 1* and *Supplementary file 1—Data S1*). We evaluated how different recombination rates and marker densities affect the quality of detected IBD segments, performed IBD caller-specific parameter optimization, and benchmarked different IBD detection methods with their optimized parameters at both the IBD segment and downstream inference levels. Furthermore, we validated our findings from simulation analysis with empirical data sets constructed from subsets of samples from the publicly available whole-genome sequencing database MalariaGEN *Pf7*. Our findings indicate that a high recombination rate (given the same mutation rate) is associated with a low SNP density (per genetic unit), which substantially affects the accuracy of the detected IBD segments. To obtain optimal results, we generally recommend using `hmmIBD` when phased genotype data from haploid genomes are available. If human-oriented IBD callers are used, we recommend optimizing the parameters prior to applying to *Pf* genomes.

## Results

### Low SNP density due to high recombination rate affects the accuracy of IBD calls

Accurate estimation of IBD segments often requires dense genetic markers to capture ancestral relationships left by recent recombination events (*Thompson, 2013*; *Browning and Browning, 2012*; *Zhou et al., 2020*; *Tang et al., 2022*; *Kelleher et al., 2019*; *Speidel et al., 2019*). However, *Pf* has very low marker densities per genetic unit, which may significantly affect the accuracy of inferred IBD segments.

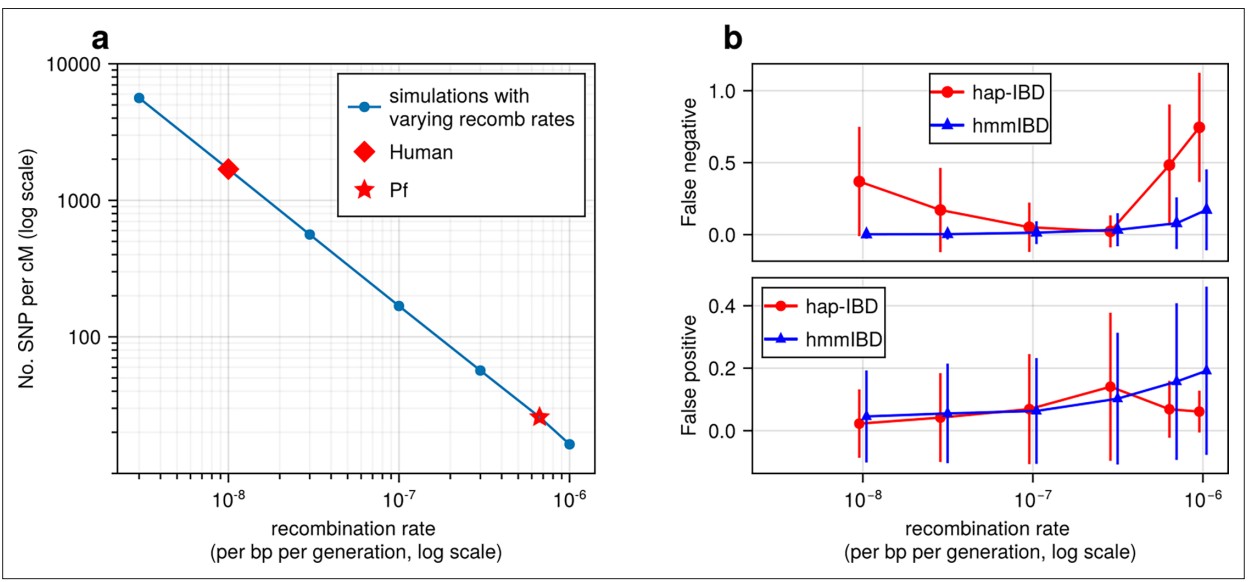

**Figure 2.** High recombination rates reduce genetic marker density and affect the quality of detected IBD segments. (**a**) The number of common single nucleotide polymorphisms (SNPs) (minor allele frequency ≥ 0.01) per genetic unit (centimorgan, cM) in simulated genomes with different recombination rates. In these simulations (blue line), the mutation rates are fixed; the recombination rates vary widely to include the rate for both humans (red diamond) and *Pf* (red star). For each recombination rate, n = 4 independent simulations of chromosomes were performed. Data were plotted in the form of mean (marker) ± standard deviation (vertical lines, which are difficult to visualize given low variation among chromosomes). (**b**) Accuracy of IBD segments detected from genomes simulated with different recombination rates. The accuracy of IBD segments is measured by the false negative rates (top panel) and false positive rates (bottom panel). The plotted error rates represent the genome-wide rates (as defined in Methods) of IBD segments identified with default IBD caller parameters unless stated otherwise in *Supplementary file 2—Table S4* . These rates are based on one representative set from n = 3 simulation sets. The plotted vertical lines indicate standard deviations of error rates across all genome pairs in the representative simulated set. Both the vertical lines and markers are horizontally staggered for clarity. Only error rates for two IBD detection methods, `hmmIBD` and `hap-IBD`, are included in (**b**) for simplicity. The error rates for all 5 IBD callers are provided in *Figure 2—figure supplement 1*. For both (**a**) and (**b**), the genomes were simulated under the single-population model (see Methods). Note that log scales are used for the *y* axis in (**a**) and the *x* axis in (**b**).

The online version of this article includes the following figure supplement(s) for figure 2:

**Figure supplement 1.** High recombination rates affect the quality of detected IBD segments (extending *Figure 2*).

To assess how recombination rates affect marker density per genetic unit and the detection of IBD segments, we simulated genomes with varying recombination rates but a fixed mutation rate. Under a single-population demographic model (see Methods), we found that the density of common biallelic SNPs (minor allele frequency ≥ 0.01) per cM, in selectively neutral scenarios, is inversely correlated with recombination rates ranging from $3 \times 10^{-9}$ to $10^{-6}$ per base pair per generation (*Figure 2a*). For instance, the SNP density of *Pf*-like genomes is 25 SNPs per cM, which is approximately 1/67 of that of the human-like genomes (1660 SNPs per cM). We further assessed how low marker densities associated with high recombination rates affect the accuracy of detected IBD segments, calculating two metrics (via *ishare/ibdutils*, see Code availability), including the false negative rate (FN), which represents the proportion of a true IBD segment (obtained via `tskibd` *Guo et al., 2024*) not covered by inferred IBD segments of the same genome pairs, and the false positive rate (FP), which indicates the fraction of an inferred segment not covered by true segments of the same genome pairs (see Methods for detailed definitions, and *Figure 1* method overview). Our analysis showed that as the recombination rate increases, both the genome-wide FN and FP (*Figure 2b*) increase for IBD inferred from `hmmIBD`. The patterns vary in the other four IBD detection methods, and all suffer elevated FNs and/or FPs as the recombination rate increases (*Figure 2b*; *Figure 2—figure supplement 1*), with the exception of `isoRelate`, which has better IBD quality with lower marker densities. The results suggest that low SNP density per genetic unit can dramatically affect the reliability of detected IBD segments.

## Varying quality of IBD inferred from simulated *Pf* genomes via different IBD callers

Multiple IBD callers have been used for *Pf*, including *Pf*-oriented, Hidden Markov Model-based methods, such as `hmmIBD` (*Schaffner et al., 2018*) and `isoRelate` (*Henden et al., 2018*), and those originally designed for human genomes, such as `Refined IBD` (*Morgan et al., 2020*) and `Beagle` (version 4.1) (*Shetty et al., 2019*). We analyzed `hap-IBD` (*Zhou et al., 2020*) and `phased IBD` (*Freyman et al., 2021*) in addition to `hmmIBD`, `isoRelate`, and `Refined IBD`, since the former represents two recent key advancements in the development of IBD detection methods that scale well to large sample sizes and genome sizes.

To evaluate the applicability and accuracy of these IBD detection methods in analyzing *Pf* genomes, we performed benchmarking analyses in simulated genomes (*Figure 1*, top panel), mimicking the high recombination rate and the decreasing population size of *Pf* populations. Our analyses include three sets of comparisons: (1) baseline benchmarking, where we mainly used the default parameter values for each IBD caller and compared the performance in *Pf* genomes at the level of an IBD segment and their simple statistics; (2) post-optimization benchmarking, where we used parameter values optimized specifically for each IBD caller so that the comparisons are based on the optimal performance of each method; (3) human-like genome benchmarking, where detected IBD segments are expected to have low error rates for human-oriented IBD callers and thus were used as an internal control to validate our benchmarking pipeline (see *Supplementary file 2—Table S1* for the IBD caller parameters and Methods for demographic models).

Baseline benchmarking analysis shows that all callers except `hmmIBD` suffer from high FN rates, especially for short IBD segments (genome-wide FN/FP rates reflect shorter IBD segments as most segments are short; *Figure 3*). Similarly, genetic relatedness metrics based on pairwise total IBD are largely underestimated for most callers (*Figure 3—figure supplement 1*). We found that by default, `hmmIBD` has relatively low FN/FP error rates (*Figure 3*) and is less biased for relatedness estimates (*Figure 3—figure supplement 1*). In contrast, `isoRelate` and human-oriented callers have high FN rates and varying FP rates (*Figure 3*). Thus, both *Pf*- and human-oriented IBD callers can suffer high error rates when detecting IBD from genomes with a high recombination rate and a low marker density, with the extent depending on underlying assumptions and methodologies.

## IBD-caller-specific parameter optimization for *Pf* improves IBD accuracy

Since these IBD callers are optimized for different species or genotype data sets, we hypothesized that optimization of IBD caller parameters under a unified framework designed for *Pf* genomes can improve the performance of these callers in analyzing malaria parasite data. As searching the entire IBD caller parameter space is inefficient, our optimization focused mainly on parameters potentially

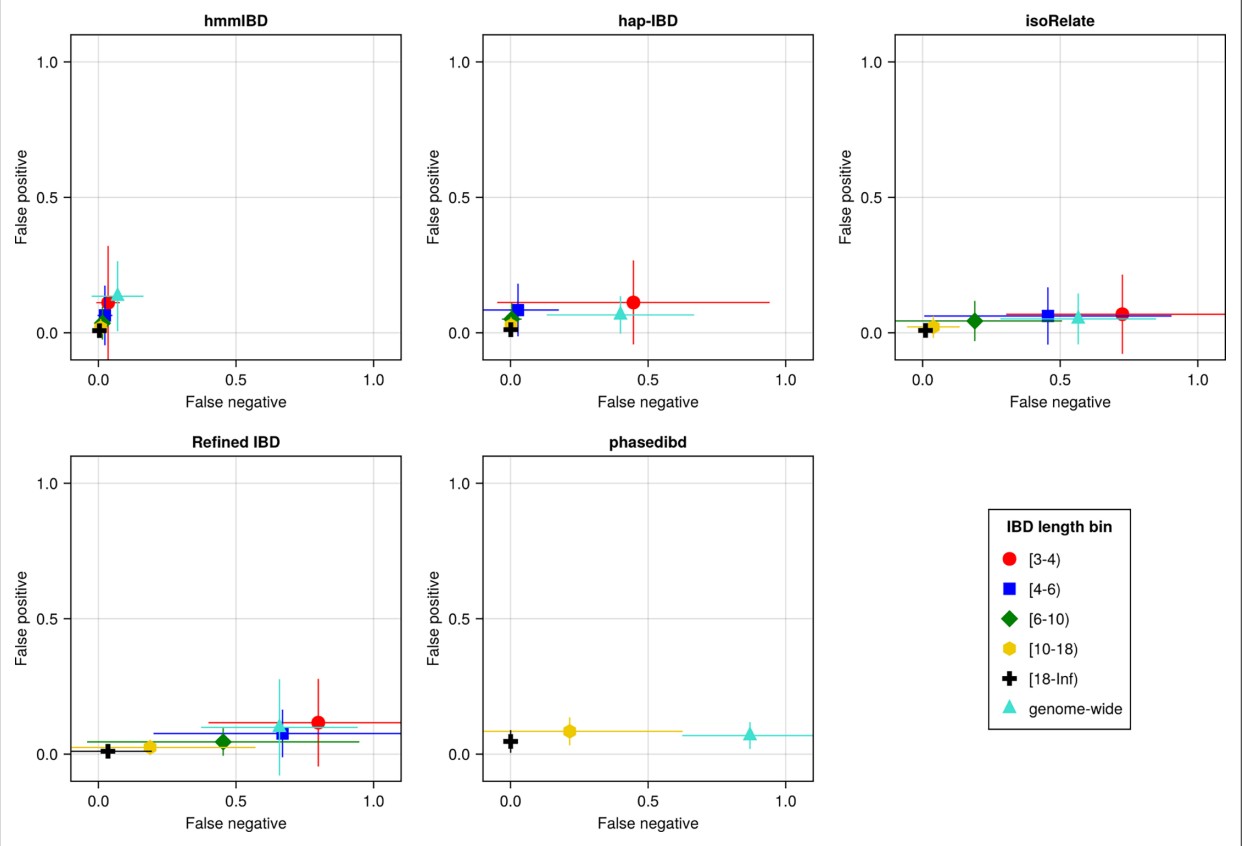

**Figure 3.** The accuracy of IBD segments detected from *Pf* genomes varies across IBD callers. IBD segments were inferred from genomes simulated under the single-population model with a shrinking population size and a recombination rate compatible with *Pf*. The accuracy of IBD was evaluated using the calculated false positive rate (*y* axis) and false negative rate (*x* axis). The rates were calculated for different length bins in centimorgans, including [3-4], [4-6], [6-10], [10-18], [18, inf] centimorgans and at the genome-wide level (defined by overlapping analysis between true IBD segments and inferred IBD segments from each genome pair). These rates are based on one representative set from n = 3 simulation sets. The plotted vertical and horizontal lines represent the standard deviations of error rates (horizontal for false negatives and vertical for false positives) calculated across all relevant segments for length-bin specific rates or across all genome pairs for genome-wide rates in the representative simulated set. The titles of the subplots indicate the IBD callers analyzed. The results of the simulations under the multiple-population model and the UK human demographic model are provided as *Supplementary file 2—Data S2*.

The online version of this article includes the following figure supplement(s) for figure 3:

**Figure supplement 1.** Pairwise total IBD of simulated Pf-like genomes from the different IBD callers compared with those based on true IBD (before parameter optimization).

affected by or needing adjustment due to differences in marker density between the high-recombining species (e.g. *Pf*) and the lower-recombining species (e.g. humans). For IBD callers that do not explicitly have marker density-related parameters, such as `hmmIBD`, we explored other parameters that likely affect IBD quality. We performed grid searches to find parameter values that generate inferred IBD with low and balanced error rates (see *Supplementary file 2—Table S2* for parameters explored and their corresponding values, and *Supplementary file 2—Data S2* for detailed results).

We found that most IBD callers have a key parameter that can dramatically affect their FN/FP rates (*Supplementary file 2—Table S2*). For example, the FN rates of IBD called from `hap-IBD` change substantially when the `min-marker` parameter varies (see *Supplementary file 2—Data S2*). With a value of 70, the FN rate for short IBD segments dramatically decreases such that the FN and FP error rates become more balanced (*Figure 4a*). Consistently, genetic relatedness estimates change from being highly underestimated before parameter optimization (*Figure 4b*, left column) to being more balanced after optimization (*Figure 4b*, right column). Similar improvements were observed for `Refined IBD` and `phased IBD` (*Figure 4—figure supplement 1*). In contrast, the quality metrics remained largely unchanged during parameter optimization attempts for `hmmIBD` and `isoRelate`,

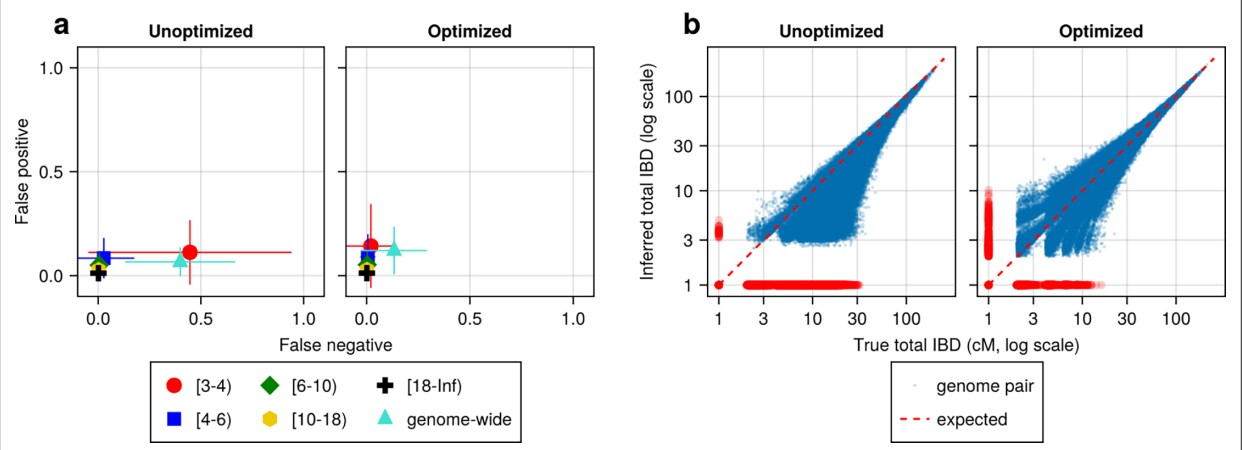

**Figure 4.** IBD caller-specific parameter optimization can improve the quality of IBD segments inferred from simulated *Pf* genomes (using hap-IBD as an example). (**a**) Quality of detected IBD measured by false positive and false negative rates before (left column) and after (right column) hap-IBD-specific parameter optimization. As indicated in the axis legend, the error rates were calculated for different length ranges (in centimorgans), including [3-4], [4-6], [6-10], [10-18], [18, inf] and at the genome-wide level. These rates are based on one representative set from n = 3 simulation sets. The plotted vertical and horizontal lines represent the standard deviations of error rates (horizontal for false negatives and vertical for false positives) calculated across all relevant segments for length-bin-specific rates or across all genome pairs for genome-wide rates in the representative simulated set. (**b**) Quality of detected IBD measured by total genome pairwise IBD, an estimate of genetic relatedness, before (left column) and after (right column) parameter optimization. Data of n = 1000 haploid genomes from a representative simulation set was plotted with each subplot for `hap-IBD` before and after parameter optimization as indicated in the titles. Each dot represents a pair of genomes with the coordinates x and y being true and inferred total IBD. In (**b**), the blue dots are the pairs with nonzero true and inferred total IBD, while red dots are pairs with either true total IBD or inferred total IBD being 0; zero-valued total IBD was replaced with 1.0 cM for visualization purposes. The red dotted line of y = x indicates the expected pattern, that is, true total IBD equal to inferred total IBD if the inferred IBD was 100% accurate. Note that log scales are used in both the x and y axes in (**b**).

The online version of this article includes the following figure supplement(s) for figure 4:

**Figure supplement 1.** IBD caller-specific parameter optimization can improve the quality of IBD segments inferred from simulated Pf genomes (extending *Figure 4*).

**Figure supplement 2.** Quality of IBD segments called via human-oriented callers from simulated genomes with UK human population demographic history but different recombination rates (human versus Pf rates).

with `hmmIBD` being more accurate and unbiased and `isoRelate` suffering from high false negative rates and underestimated relatedness (*Figure 4—figure supplement 1*).

The human-oriented IBD callers, when not optimized for *Pf*, underperformed `hmmIBD`, especially for `Refined IBD` and `phased IBD` (*Figure 4—figure supplement 1a*). To exclude potential problems in our benchmarking pipeline, we simulated genomes with recombination rates and demographic history consistent with the human population in the UK (*Figure 4—figure supplement 2a*, left column, and *Figure 4—figure supplement 2b*, left column; also see Methods). These callers, evaluated *without Pf*-optimized parameters, indeed perform much better on human genomes, showing consistently lower FN/FP error rates (*Figure 4—figure supplement 2a*, left column) and less biased total IBD-based relatedness estimates (*Figure 4—figure supplement 2b*, left column), compared to *Pf*-like genomes (*Figure 4—figure supplement 1a and b*, left columns). The results support the robustness of our benchmarking pipeline (*Figure 4—figure supplement 2a and b*, left columns) and demonstrate challenges in applying human-oriented IBD callers to *Pf*. Furthermore, we found that IBD caller performance varies with demographic configurations (the single-population model in *Figure 4—figure supplement 1a and b*, left columns, versus UK human model in *Figure 4—figure supplement 2a and b*, right columns) even with the *same* (*Pf*) recombination/mutation rates and default IBD caller parameters, suggesting that the optimization is demography-dependent (see details in *Supplementary file 2—Data S2*).

## Post-optimization benchmarking via downstream inferences

IBD-based downstream analyses, such as estimation of $N_e$, selection signals, and population structure, are key applications of IBD segments in population genetics, which often rely on the high quality of

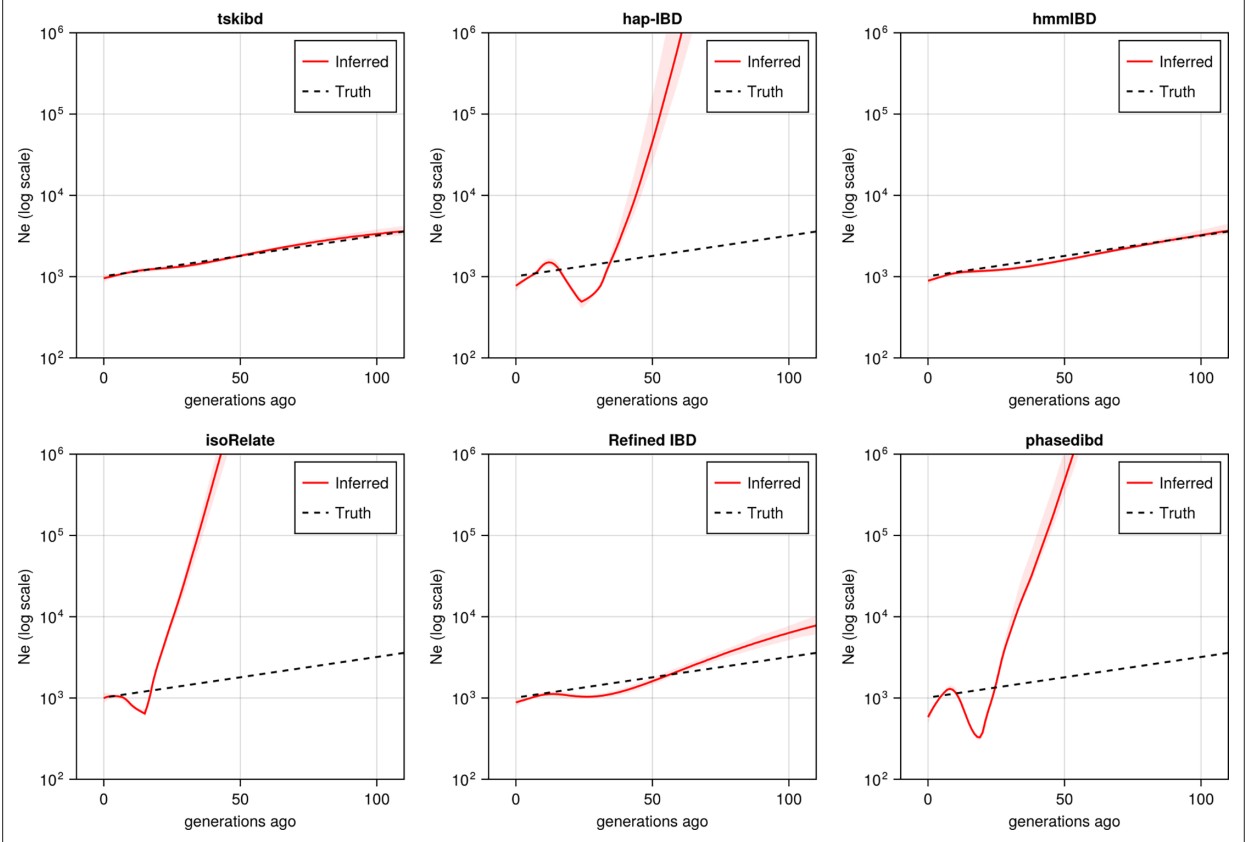

**Figure 5.** Post-optimization benchmarking of different IBD callers by comparing downstream estimates $N_e$. With parameters optimized for each IBD caller, the performance of IBD callers was evaluated by comparing the $N_e$ trajectory for the recent 100 generations estimated via `IBDNe` based on true (black dashed line) IBD versus inferred IBD (red solid line). True IBD was calculated from simulated genealogical trees via `tskibd`; inferred IBD includes those inferred from `hap-IBD`, `hmmIBD`, `isoRelate`, `Refined IBD`, and `phased IBD`, with their $N_e$ estimates shown from top left to bottom right. The shaded areas surrounding the red lines indicate 95% confidence intervals as determined by `IBDNe`. The plots show results from one representative set of n=3 replicated simulation sets. See *Figure 5—figure supplement 5* for pre-optimization results. Note that log scales are used on the y axes.

The online version of this article includes the following figure supplement(s) for figure 5:

**Figure supplement 1.** Post-optimization benchmarking of different IBD callers by comparing downstream estimates of selected loci and population structure.

**Figure supplement 2.** Population-level total IBD per length bin.

**Figure supplement 3.** The effect of including and excluding short IBD segments (<4 cM) on estimation.

**Figure supplement 4.** Pre-optimization benchmarking of different IBD callers by comparing downstream estimates of selected loci and population structure (for comparison with *Figure 5—figure supplement 1*).

**Figure supplement 5.** Pre-optimization benchmarking of different IBD callers by comparing downstream estimates of $N_e$ (for comparison with *Figure 5*).

input IBD segments. With optimized IBD caller-specific parameters tailored for *Pf*-like genomes, we can expect IBD callers to perform at their best, which allows high-level benchmarking by comparing IBD-based downstream estimates (*Figure 5* and supplements).

For IBD-based selection detection, we simulated positive selection on each of the 14 chromosomes via the single-population model and identified IBD peaks with different callers (see Methods). These peaks, inferred as regions under selection, were considered true signals if they contained the selected site from simulations. We found that most callers can capture the majority of the simulated signals except `Refined IBD`, which is less sensitive and only detects 2–3 out of 14 selected loci (*Figure 5— figure supplement 1a*); `isoRelate` shows an increased level of false positives or low signal-to-noise ratios, evident in IBD coverage curves (*Figure 5—figure supplement 1a*).

For IBD-based population structure inference, we simulated *Pf*-like genomes under a selectively neutral condition using the multiple-population demographic model and performed IBD network community detection via InfoMap (*Rosvall et al., 2009*; *Csardi and Nepusz, 2006*) to define subpopulations (see Methods). Similarly to IBD-based selection signal detection, we found that IBD inferred from most callers can accurately recapitulate the simulated population structure, comparable to true IBD (*Figure 5—figure supplement 1b*). The exception is that `isoRelate` tends to generate many smaller groups, which is likely due to high FN rates for short IBD segments, missing connectivity among distantly related genomes, and only showing closely related subgroups of small sizes. As indicated by the low adjusted Rand indices, there was little agreement between the community labels inferred by `isoRelate` and the true labels.

For IBD-based $N_e$ inference, we simulated neutral *Pf* genomes using the single-population model and estimated $N_e$ from detected IBD via `IBDNe` (*Browning and Browning, 2015*). We found most of the compared IBD callers suffer from wild oscillations, which have previously been observed (*Browning and Browning, 2015*), and deviate significantly from the truth for older generations (*Figure 5*), consistent with the general pattern of high error rates in shorter IBD length bins for these IBD callers (*Figure 4—figure supplement 1a*, right column). Meanwhile, the IBD inferred by `hmmIBD` generated highly accurate estimates comparable to true IBD (*Figure 5*). We explored the mechanisms underlying bias in $N_e$ estimates and found that the strong bias is likely due to the underestimation of population-level total IBD for short IBD segments, which is most obvious for `hap-IBD`, `isoRelate`, and `phased IBD`, followed by `Refined IBD` (*Figure 5—figure supplement 2*). For `hmmIBD`, both the estimates of $N_e$ (*Figure 5*) and population total IBD are relatively unbiased (*Figure 5—figure supplement 2*, column 2), consistent with its relatively low and balanced FP/FN rates (*Figure 3*, leftmost panel). These results suggest that IBD-based $N_e$ estimates are highly sensitive to the quality of input IBD segments, and that `hmmIBD` is more accurate for this analysis.

Given that $N_e$ estimation in *Pf* is very sensitive to the quality of detected IBD, we explored whether excluding short, error-prone IBD segments (< 4 cM) could improve $N_e$ estimates for callers other than `hmmIBD`. The exclusion results in reduced oscillation of the trajectory for some callers, like `hap-IBD`, but wide confidence intervals or underestimation in older generations, in both simulated and empirical data (*Figure 5—figure supplement 3*). We then explored reasons underlying the recent oscillation or drop (around 20 generations ago) commonly observed in the estimated $N_e$ trajectories (*Figure 5—figure supplement 3a* second row and *Figure 5—figure supplement 3b*, both rows) (*Morgan et al., 2020*; *Browning and Browning, 2011*; *Harris et al., 2020*). We hypothesized that this oscillation is partially due to IBD segments with TMRCA < 1.5 generations ago being included in the `IBDNe` input (*Browning and Browning, 2015*). We found that removing these segments can greatly mitigate this problem (*Figure 5—figure supplement 3a*), especially for `hmmIBD` and `Refined IBD`. The findings suggest caution when interpreting (1) a recent drop in an estimated $N_e$ trajectory in empirical data sets where TMRCA-based filtering is less practical, and (2) extremely large estimates in older generations stemming from high error rates for short IBD segments.

To confirm that parameter optimization improves downstream inferences, we compared post-optimization results (*Figure 5—figure supplement 1* and *Figure 5*) with pre-optimization estimates (*Figure 5—figure supplement 4* and *Figure 5—figure supplement 5*). We found that parameter optimization increased the accuracy of selection detection (in `isoRelate`, `Refined IBD`, and `phased IBD`), improved population structure inference (`phased IBD` and `hap-IBD`), and reduced oscillation on the $N_e$ trajectory (`Refined IBD`).

## Validation in empirical data sets

We further validated the findings from simulation analysis in empirical data sets, by constructing 'single' or 'multiple' population data sets based on the MalariaGEN *Pf*7 data (*Abdel Hamid et al., 2023*; see Methods for details). As true IBD segments are not available here, we focused on high-level benchmarking by evaluating whether IBD-based downstream estimates are consistent with expected patterns, including $N_e$ estimation and selection signal detection with 'single' population data sets and InfoMap population structure inference with the 'multiple' population data set (*Supplementary file 2—Table S3-S5*).

With optimized parameters, all callers, except `Refined IBD`, capture most known selection signals from the Southeast Asia data set. These signals include selective sweeps associated with antimalarial

drug resistance and sexual commitment, such as dihydrofolate reductase (*dhfr*) (*Miotto et al., 2013*), multidrug resistance protein 1 (*pfmdr1*) (*Koenderink et al., 2010*), amino acid transporter 1 (*pfaat1*) (*Amambua-Ngwa et al., 2019*), chloroquine resistance transporter (*pfcrt*) (*Martin and Kirk, 2004*), dihydropteroate synthase (*dhps*) (*Brooks et al., 1994*), Apicomplexan-specific ApiAP2-g (*ap2-g*) (*Early et al., 2022*) and apicoplast ribosomal protein S10 (*arps10*) (*Miotto et al., 2015*; *Figure 6a*). hmmIBD detects more peaks but suffers from noise, likely due to the relatively high FP to FN ratios for short (<4 cM) IBD segments (*Figure 2* and *Figure 3*).

Similar to the simulation analysis, IBD detected using most callers resulted in $N_e$ estimates with unrealistic oscillations for the Southeast Asia data set, including extremely large estimates in the more distant past (> 20 generations ago; *Figure 6b*). The problems are much less severe with the estimates from hmmIBD, which mirrored the expected reduction in malaria in this region due to the intense efforts to eliminate malaria in recent decades (*World Health Organization, 2024*).

InfoMap-based community detection reveals expected continental population structure across most callers: African *Pf* parasites are less structured, and Southeast Asian parasites are more structured and distinct from Oceanian parasites (*Figure 6c*), consistent with previous non-IBD-based methods (*Abdel Hamid et al., 2023*). isoRelate IBD estimates generate many small, close groups, likely due to high false negative rates for short IBD segments, especially in high transmission settings like Africa where parasites have low relatedness and mainly share short IBD segments (*Figure 6c*).

To further confirm the improvement of IBD-based estimates through parameter optimization, we performed analyses with IBD detected with unoptimized parameters (see *Supplementary file 2—Table S1*). We found that the height and number of IBD peaks for hap-IBD and phased IBD decreased significantly (*Figure 6—figure supplement 1 versus Figure 6a*). Pre-optimization $N_e$ estimates show more extreme oscillations, especially for human-oriented IBD callers (*Figure 6—figure supplement 1b*). Pre-optimization IBD estimates from hap-IBD and phased IBD fail to reveal the expected population structure, particularly in African parasite populations (*Figure 6—figure supplement 1c*). These differences underscore the importance of parameter optimization for *Pf*, especially for IBD callers not validated for *Pf*.

## Computational efficiency comparison

With the decrease in whole-genome sequencing cost and the increase in sample availability, it is important to prioritize IBD callers that scale well for large sample sizes, such as MalariaGEN *Pf*7 (n = 20,864; *Abdel Hamid et al., 2023*). We compared the IBD inference time and maximum memory usage for different IBD callers with or without parallelization. When using a single thread, probabilistic inference algorithms like isoRelate, Refined IBD, and hmmIBD are about two orders of magnitude slower than those based on identity-by-state-based or positional Burrows-Wheeler transform (PBWT) based algorithms, such as hap-IBD and phased IBD (*Figure 7a*). Maximum memory consumption is highest in Refined IBD, with hmmIBD being around 10 times more efficient (*Figure 7—figure supplement 1*). With multithreading, the patterns are similar to single-thread comparison as most allow parallelization (*Figure 7b* and *Figure 7—figure supplement 1b*). The exception is hmmIBD, as it currently only supports single-thread computation. Despite the computational efficiency of hmmIBD compared to isoRelate and its high accuracy in detecting IBD segments from *Pf* genomes, it remains significantly slower than IBS-based methods, highlighting the need for further enhancements for large data sets like MalariaGEN *Pf*7.

## Discussion

In this work, we evaluated the reliability of IBD detection methods in high-recombining genomes of malaria parasites. Our findings indicate that low marker density per genetic distance significantly affects IBD detection accuracy. Optimizing parameters of IBD detection methods for *Plasmodium* genomes enhances the accuracy of detected IBD segments, thereby improving subsequent downstream analyses such as the inference of positive selection signals and population structures. However, variations in performance among IBD detection methods remain substantial, particularly in analyses sensitive to IBD quality, such as the estimation of effective population size ($N_e$), even after parameter optimization. We generally recommend hmmIBD for IBD estimation in *Plasmodium* genomes when phased genotype data of haploid genomes are available. We further emphasize the necessity

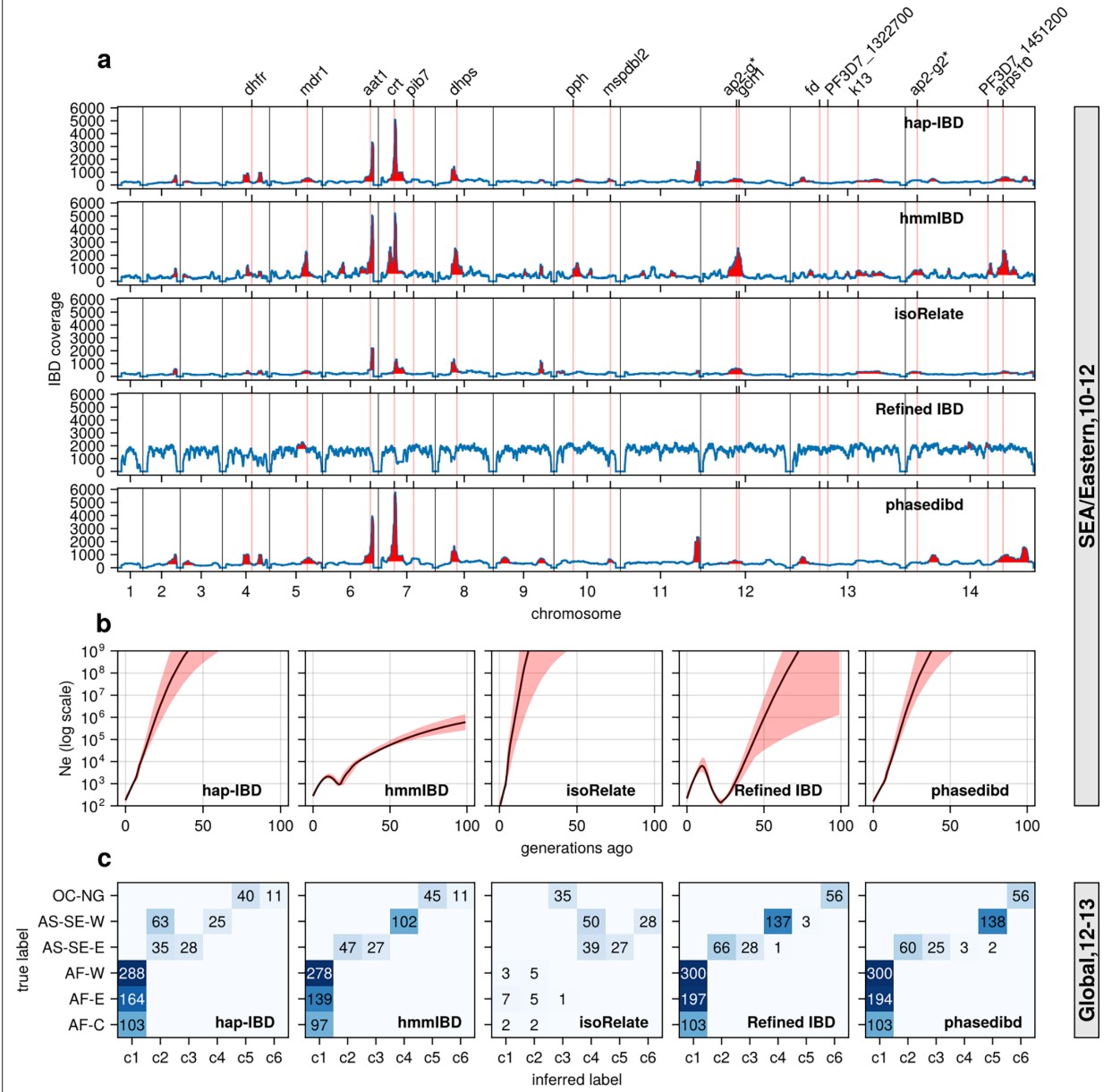

**Figure 6.** Validation of the performance of IBD callers in empirical data sets by comparing IBD-based downstream analyses. (**a**) IBD coverage and detected selection signals in the SEA data set using different IBD callers (rows 1 to 5). Annotations and corresponding vertical dotted lines at the top indicate the center of known and putative drug resistance genes and genes related to sexual commitment; red shading indicates regions that are inferred to be under positive selection (see Methods for definitions). (**b**) $N_e$ estimates of the SEA data set based on IBD inferred from different callers. Line plots are point estimates; the shaded areas around the line plots indicate confidence intervals based on bootstrapping (generated by `IBDNe`). (**c**) Inference of the population structure of the structured data set by the InfoMap community detection algorithm using the IBD inferred from different IBD callers. The rows of the heatmap are geographic regions of isolates, and the columns are the largest, inferred communities, labeled as c1 to c6. The heatmap color represents the number of isolates in each block with the given row and column labels. The columns are rearranged so that the diagonal blocks tend to have the largest values per row for better visualization. Note that log scales are used in y axes in (**b**).

The online version of this article includes the following figure supplement(s) for figure 6:

**Figure supplement 1.** Performance of IBD callers with empirical data sets before parameter optimization (in comparison with post-optimization performance shown in *Figure 6*).

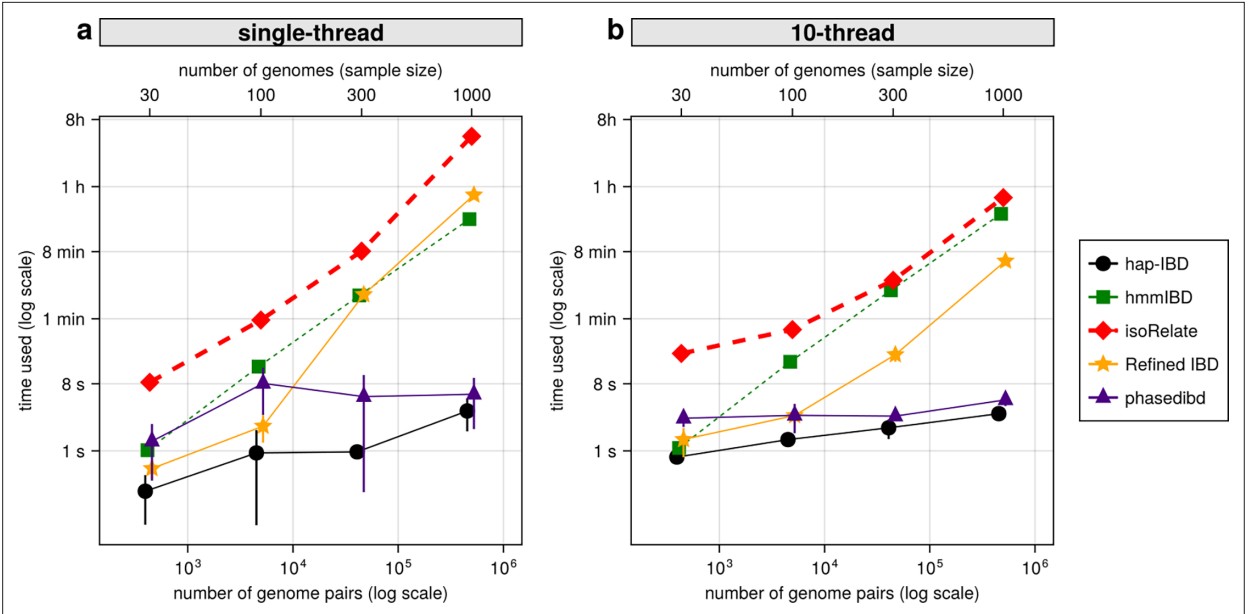

**Figure 7.** Comparison of computational runtime for IBD calling process for different callers. (**a**) Runtime for different IBD callers to detect IBD from genomes of different sample sizes in single-thread mode. The comparison is based on *Pf* genomes of size 100 cM simulated under the single-population model. The *x*-axis tick labels include the number of pairs of genomes simulated and analyzed (below the plot, reflecting the number of computation units) and the number of haploid genomes (above the plot, representing sample size) analyzed. The line styles and markers for different callers/tools are provided in the legend box on the far right of the figure, which is shared across the two subplots. Values on the *y* axis represent means (markers) and standard deviations (vertical error lines) from n = 3 sets of independent simulations. Note that at each sample size, the *x* values for different IBD callers are slightly staggered to prevent the error lines from overlapping; some of the error lines are hard to visualize as they are relatively small. (**b**) Runtime in multithreading mode. (**b**) is organized similarly to (**a**) except that the IBD calling processes were run in multithreading mode with 10 CPU threads. Note that log scales are used in *y* axes and bottom *x* axes. Also, see *Figure 7—figure supplement 1* for the maximum memory usage for different callers.

The online version of this article includes the following figure supplement(s) for figure 7:

**Figure supplement 1.** Comparison of maximum computer memory used for IBD calling process for different callers.

of performance evaluation and parameter optimization for IBD callers prior to application in untested species or scenarios.

Comparing the performance of multiple IBD detection methods requires a unified framework, which should include a uniform definition of accuracy and a simulated ground truth that mimics *Pf* genomes. Our benchmarking framework incorporates several novel features. First, it utilizes a consistent definition of IBD length-specific accuracy based on the overlap of IBD segment lengths, closely aligned with metrics used in `hap-IBD` (*Zhou et al., 2020*), `phased IBD` (*Freyman et al., 2021*), and `Refined IBD` (*Browning and Browning, 2013*), as detailed in our Methods section. The approach differs from the original evaluation of `hmmIBD`, which assesses the accuracy of IBD based on the fraction of SNPs that share IBD, which could overlook the precise accuracy of the length of the detected IBD segments (*Schaffner et al., 2018*). In particular, the original study of `isoRelate` defined the accuracy (true positive rate) using a less stringent overlap-by-count criterion where a segment is counted as accurate when at least 50% of a detected IBD segment is overlapped by true segments (*Henden et al., 2018*). Second, we generated *Pf*-like genomes via population genetic simulation that reflect a realistic distribution of IBD segment lengths. This contrasts with previous studies, where methods such as `hap-IBD` and `Refined IBD` used human-like data from population genetic simulations (*Browning and Browning, 2013*; *Zhou et al., 2020*), whereas `hmmIBD`, `isoRelate`, and `phased IBD` relied on simulations based on artificial recombination or pedigree models (*Henden et al., 2018*; *Schaffner et al., 2018*). These models (non-population-based genetic simulation) often produce long shared IBD segments typical of close relatives, failing to capture the IBD length distribution in population samples predominantly comprising distant relatives. Third, our benchmarking extends beyond the segment-level evaluation of IBD callers and includes downstream inferences of population structure

and effective population size, providing a more thorough assessment of their application in real-world analyses.

Of note, our benchmarking highlights the high false negative rate of short segments detected by `isoRelate`, despite it being developed for malaria parasites. The relatively poor performance of `isoRelate` compared to `hmmIBD` is likely due to differences in the underlying HMM models, where the `isoRelate` model assumes unphased data even when phase is provided in our benchmarking analysis. Given that `isoRelate` can be applied directly to unphased genotype data, it might outperform `hmmIBD` when genotype phasing, required by `hmmIBD`, is error-prone. However, this possibility was not examined in this study and warrants further investigation beyond the current scope.

The density of markers per genetic unit plays a crucial role in IBD detection. IBS-based methods, such as `hap-IBD` and `phased IBD`, first identify long IBS segments (≥ 2 centimorgan) as candidate IBD segments, and subsequently merge short ones separated by small gaps (*Zhou et al., 2020*), allow a certain number of discordant markers to account for phasing errors (*Freyman et al., 2021*), or eliminate false positive segments by removing candidate segments supported by only a small number of markers (*Zhou et al., 2020*). Similarly, `Refined IBD`, which combines an IBS-based method with an HMM probabilistic model, uses a LOD score to decide whether a candidate IBD segment should be rejected or accepted (*Browning and Browning, 2013*). In these studies, default values for marker density-related parameters were shown to be effective for human genomes but have not been evaluated in high-recombining genomes. We evaluated how different levels of per-genetic-unit marker density affect the detection of IBD segments by varying recombination rates in simulations. With simulated *Pf*-like genomes of low marker density and high recombination rate, we found that human-oriented IBD callers suffer high false negative rates. One potential explanation is that the thresholds optimized for human data are too stringent for *Pf*, causing excessive rejection of candidate segments. The effect of marker density on IBD detection is further confirmed by our findings that adjusting the values of marker density-related parameters using a grid-search approach could significantly reduce IBD error rates and generate more accurate IBD-based downstream estimates. Even though IBD accuracy can be improved by parameter optimization, we found that error rates of the detected IBD segments are still higher in *Pf* genomes than those of human genomes even after IBD caller parameters are optimized. There are several possible explanations for the high error rates of IBD segments detected from low marker density data: (1) Detected IBD segments can only start and end at the genotype marker site, which may not reflect true IBD end points; (2) A lower marker density is linked with greater uncertainty in the distribution of IBD endpoints *Browning and Browning, 2020*; (3) Ancestral relationships, including IBD, are too difficult to be reliably inferred given limited mutational information (*Ishigohoka and Liedvogel, 2025*; *Speidel et al., 2019*; *Kelleher et al., 2019*; *Mehra et al., 2025*).

In our benchmark analysis, we used only common biallelic SNPs as markers for inferring IBD, excluding rare variants and indels. The use of this additional information can potentially provide denser genotype information, thus enhancing our understanding of the population's ancestral relationships, a key aspect on which the inference of the IBD segment depends. For instance, large-scale whole-genome sequencing studies reveal that rare variants account for the majority of all segregating sites (*Ahouidi et al., 2021*; *Taliun et al., 2021*), which contain crucial information for deciphering recent evolutionary history. However, rare variants are typically not utilized for two main reasons: (1) rare genotypes are very sparsely distributed across many sites and are less informative per site (*Browning and Browning, 2020*; *Schaffner et al., 2018*); (2) rare genotype calls are more prone to genotyping or phasing errors. As a result, including rare variation may cause reduced accuracy in detected IBD segments due to genotyping/phasing errors and increase IBD inference time due to marker density. Although our simulation analysis indicates that including rare variants is of little effect or detrimental for IBD detection (*Supplementary file 2—Data S2*), the findings could be skewed due to the absence of genotyping errors, the small sample size simulated, and the limited number of cut-off values tested. Further investigation in the context of large sample sizes and varying levels of genotype errors is needed to inform the usability of rare variants for different IBD detection algorithms. Indels are another significant source of underutilized genetic variations in the *Pf* genome, with abundance on par with that of biallelic SNPs in the MalariaGEN *Pf*7 data set (*Abdel Hamid et al., 2023*). These indels are, in part, the result of microplasticity related to the high AT content (up to 90% in non-coding regions) in *Pf* genomes

(*Hamilton et al., 2017*). Using these variants could also increase the marker density to infer IBD segments. However, additional research is necessary to determine whether the inclusion of these variants can reduce uncertainty in the inference of IBD for *Pf* or introduce more bias due to challenges such as sequence-read mapping.

While demonstrating high accuracy in IBD detection and downstream analysis, `hmmIBD` tends to be slower than IBS-based methods like `hap-IBD`. The trade-off between speed and accuracy may limit `hmmIBD`'s suitability for large sample-size data sets. Our ongoing efforts are directed towards enhancing the computational efficiency and functionalities of the model used by `hmmIBD` in an adapted tool, thereby extending its applicability to large-scale data sets of relatively small and high-recombining genomes, such as *Plasmodium* genomes.

Although a substantial portion of this study concentrated on *Pf*, the main findings and methodologies may be relevant to high-recombining species beyond *Pf*. For instance, in regions with intermediate and low malaria transmission, the incidence of *Pf* has markedly decreased, allowing other species, such as *Plasmodium vivax*, to become predominant (*World Health Organization, 2024*). Clinical malaria caused by simian *Plasmodium* species, for example *Plasmodium knowlesi*, has also increased in some geographic areas where human *Plasmodium* species have declined (*Amir et al., 2018*). We expect that the performance of IBD callers will be similar in other *Plasmodium* species, given their likely comparable high recombination rates (*Bright et al., 2014*; *Ibrahim et al., 2023*); however, the generalization would need further exploration as part of future work, considering variations in evolutionary histories and parasite biology (*Escalante et al., 2004*; *Lee et al., 2011*; *Loy et al., 2017*). Beyond *Plasmodium* parasites, there are many other high-recombining organisms (*Stapley et al., 2017*) such as Apicomplexan species like *Theileria* (*Sivakumar et al., 2014*), insects like *Apis mellifera* (honeybee; *Kent et al., 2012*; *Leroy et al., 2024*), and fungi like *Saccharomyces cerevisiae* (Baker's yeast; *Barton et al., 2008*; *Peter et al., 2018*). For these species, our optimized parameters may not be directly applicable, but the benchmarking framework established in this study can be utilized to prioritize and optimize IBD detection methods in a context-specific manner.

While we have conducted numerous simulation analyses complemented by carefully designed validation studies, our work is subject to a few caveats: (1) Our simulations did not explicitly incorporate inbreeding within the complicated life cycle of the parasite (*Anderson et al., 2000*), except for the increased inbreeding potential due to the reduced population size in the single-population model. Inbreeding can be pervasive, especially in low transmission settings, leading to a change in the length distribution of IBD toward longer segments and a potentially reduced marker density. (2) Our optimization is based on simple accuracy metrics and only focuses on a subset of parameters to allow faster iteration over different values. Investigating a larger parameter space, including genotyping error rate and higher-level accuracy metrics, may generate different optimal values and further improve IBD-based downstream estimates. (3) We assume a constant recombination rate and mutation rate as static genomic/population parameters, rather than traits capable of evolving over time. If this assumption proves to be inaccurate, such as with recombination rates that vary between individuals and populations (*Smukowski and Noor, 2011*), a more complex benchmarking framework will be required.

In conclusion, we evaluated the performance of existing IBD segment detection methods for analyzing genomic data of the malaria parasite *Pf*, which is characterized by high recombination rates and low marker densities. Our findings underscore that a high recombination rate, relative to the mutation rate, can compromise the accuracy of detected IBD segments when using methods originally calibrated for the human genome, characterized by a significantly lower recombination rate and a higher marker density (per genetic unit). The accuracy of IBD detection can be improved by parameter optimization via grid search techniques. We advocate for a context-specific evaluation of IBD detection methods when applying them to untested species. Specifically for *Pf*, our research indicates that hidden Markov model-based probabilistic methods, such as `hmmIBD`, produce less biased IBD estimates, leading to more accurate downstream inferences. This is especially important for analyses that heavily rely on the accuracy of detected IBD segments, such as $N_e$ inference. These findings will improve the accuracy of IBD detection and downstream analysis, providing more robust estimates of malaria transmission patterns, essential to effective malaria control and elimination efforts.

## Methods

### Simulation overview

We used population genetic simulations to allow the generation of (1) ground truth, including true IBD segments, true sites under positive selection, true trajectory of population size, and true sub-population assignments (population structure), and (2) inferred patterns, including IBD inferred from phased genotype data via different IBD callers and IBD-based downstream inferences of $N_e$, positive selection, and population structure. By comparing inferred patterns with ground truth, we calculated metrics at the IBD segment level and the IBD-based downstream estimate level (high level) for benchmarking and optimizing various IBD detection methods for *Pf* genomes. As described in our accompanying work (*Guo et al., 2024*), we combined the flexible forward simulator `SLiM` (*Haller and Messer, 2019*; *Haller et al., 2019*) and the efficient coalescent simulator `msprime` (*Baumdicker et al., 2022*) to simulate genomes similar to *Pf*, reflecting the high recombination rate, strong positive selection, and population size shrinkage due to malaria reduction (*Figure 1*). Detailed simulation parameter values used are provided in *Supplementary file 1—Data S1*. A detailed implementation of the simulations can be found in a dedicated GitHub repository (https://github.com/bguo068/bmibdcaller_simulations, copy archived at *Guo, 2025a*).

In these simulations, we assumed constant recombination rates over the genome, such as $6.67 \times 10^{-7}$ per base pair per generation for *Pf* (*Amambua-Ngwa et al., 2019*; *Conway et al., 1999*) and $1.0 \times 10^{-8}$ for humans (*Kong et al., 2002*), and a mutation rate of $1.0 \times 10^{-8}$ for both *Pf* (*Camponovo et al., 2023*; *Bopp et al., 2013*) and humans unless otherwise specified. Parameters for modeling population size changes, population structure, and positive selection are detailed in the following section or our related publication (*Guo et al., 2024*).

### Simulated demographic models

We used three different demographic models in the simulations, including the single-population model, the multiple-population model, and the UK European human population model (*Zhou et al., 2020*).

Single- and multiple-population models have been described in our accompanying work (*Guo et al., 2024*). The single-population model mimics malaria reduction in settings like Southeast Asia, with a population size decreasing from 10,000 to 1000 over the last 200 generations. This model was used to benchmark IBD detection methods at the IBD segment level and the (high) level of downstream estimation, including selection signal detection and $N_e$ estimation. The multiple-population model was mainly used to benchmark IBD calling methods via IBD network-based community detection. Implementation of the two models is provided in a GitHub repository (see Code availability).

The UK human demographic model, similar to the one used in *Zhou et al., 2020*, simulates a population bottleneck event from a constant size of 10,000 to 3000 that occurred 5000 generations ago, followed by growth at rates of 1.4% and 25% per generation beginning 300 and 10 generations ago, respectively. We simulated 14 chromosomes with a size of 60 cM each for a smaller genome size to reduce simulation time. This demographic model serves as a control to detect IBD segments with human-oriented callers, which can help validate our IBD accuracy evaluation pipeline. We also used this model to test whether demographic models impact the performance of IBD callers by replacing the human recombination rate with that of *Pf*.

To separate the effects of positive selection from those of demographic models and recombination rates, we mostly simulated neutral genomes by setting the selection coefficient *S* to 0 for the above models, except in the case where we needed to benchmark IBD callers for detecting positive selection signals.

### Positive selection simulation

To evaluate the performance of IBD callers via IBD-based selection signal detection, we simulated positive selection within the single-population model with selection coefficients *s* of 0.2 with a single origin starting 80 generations ago. Fourteen chromosomes with a size of 100 cM were simulated independently, each with a selected site at 33.3 cM from their left ends. For selection simulation, we conditioned on the establishment of the selective sweeps; that is, the allele under selection should not be lost in the present-day generation. If lost, the simulation was rerun for a maximum of 100 times until the selective sweep was established.

## IBD calling and default parameters

We generated true IBD from simulated genealogical trees in the tree sequences using the `tskibd` algorithm (*Guo et al., 2024*). Briefly, we sampled local/marginal trees along a chromosome and tracked changes in the most recent common ancestor (MRCA) for each pair of sample nodes. If the MRCA changes, the shared ancestral segment breaks. We report a shared ancestral segment as an IBD segment if its length exceeds a threshold, such as 2 cM.

For inferred IBD, we used phased genotype data as input. Only biallelic sites with a minor allele frequency no less than 0.01 were included, unless otherwise stated. When needed, a genetic map is generated based on the constant recombination rate as specified in the corresponding simulations. For IBD detection methods designed for diploids, we converted each haploid to a pseudo-homozygous diploid. The resulting IBD segments of each pair of pseudo-homozygous diploids (A1/A2 and B1/B2) have redundant information due to the 100% runs of homozygosity. We only keep one pair, A1-B1, and remove other combinations.

As each IBD detection method provides multiple tunable parameters, we detailed values used in *Supplementary file 2—Table S1* for both default and optimized scenarios. For the default scenarios, the parameters mostly follow the original documentation. Exceptions are parameters that need consistency across different IBD callers for benchmarking, including the minimum IBD length and the minimum minor allele frequency. The process of obtaining optimized parameter values is described below.

## Benchmarking metrics at the IBD segment level

Several metrics were calculated to benchmark IBD methods at the IBD segment level or using their simple aggregates, including the false negative rate and false positive rate, pairwise total IBD, and population-level total IBD per length bin.

The false positive rate and the false negative rate were obtained following the definition used in the work of *Zhou et al., 2020*. Rates were first calculated per segment via segment-overlapping analysis; then, they were averaged over all segments of the same length bin. The false positive rate per segment is defined as the proportion of a given IBD segment of some genome pair from the inferred set (for example, IBD called via `hmmIBD`) that is not covered by any IBD segment from the truth set (generated by `tskibd`), for the same genome pair. Similarly, the false negative rate per segment is defined as the proportion of a given IBD segment of some genome pair from the truth set that is not covered by any IBD segment from the inferred set. The average false positive rate is calculated as the average per-segment false positive rates for all inferred IBD segments the length of which falls in a certain range (length bin); the average false negative rate is calculated as the average per-segment false negative rates for all true segments of a length bin. The following length bins were used: [3-4), [4-6), [6-10), [10-18), [18, inf) cM, similar to Zhou et al. method.

Genome-wide FP/FN rates per pair and their averages across all genome pairs were calculated to capture genome-wide bias. The per-pair genome-wide false positive rate is the ratio of two sums: (1) the numerator sum is the total length of parts of all inferred IBD segments of a certain genome pair that are not covered by any true IBD segments of the same genome pair; (2) the denominator sum is the total length of all inferred IBD segments of a certain genome pair. The per-pair genome-wide false negative rate is defined in a similar way as the percentage of pairwise total true IBD that is not overlapped by any inferred segments of the same pair. We then obtain the aggregate metrics by averaging these rates for all genome pairs.

Pairwise total IBD from truth *versus* inferred set was calculated as it's a useful metric to estimate genetic relatedness and build IBD sharing networks. It was calculated as the sum of the lengths of all inferred or true IBD segments of each genome pair.

Given that the $N_e$ estimator `IBDNe` internally utilizes quantities of population-level total IBD of different length bins, these quantities were calculated here to better examine IBD accuracy for $N_e$ inference. We defined non-overlapping length bins of 0.05 cM width that cover all possible lengths. For each length bin, a population total IBD was defined as the sum of the length of all IBD segments with segment lengths falling into this bin from any genome pair.

To expedite IBD segment-level analysis and alleviate computational burdens, we developed an open-source tool, *ishare/ibdutils* (available at https://github.com/bguo068/ishare, copy archived at *Guo, 2025b*), which harnesses algorithms such as interval trees to efficiently calculate metrics like those described above.

## IBD caller parameter optimization

We optimized key IBD caller-specific parameters by iterating each parameter over the list of discrete values, or two or more parameters over a grid of discrete values. Many optimized parameters were related to marker density for IBD callers `hap-IBD`, `phased IBD`, `Refined IBD`, and `isoRelate`. Other parameters were searched to see if they potentially have a great impact on the quality of the detected IBD. The optimal values for explored parameters are determined by the length-bin-specific or genome-wide error rates (FN and FP) for detected IBD segments as defined above. The parameter value or combination of values that generates lower and generally balanced error rates was selected as optimal values. The parameters searched, the value lists explored, and the optimal values selected are summarized in *Supplementary file 2—Tables S1 and S2*. When the optimized values vary across different demographic models, we used the ones optimized for the single-population model for downstream analyses. We provided detailed simulation and IBD calling parameter values in *Supplementary file 1—Data S1* and heatmaps of error rates for all demographic models tested in *Supplementary file 2—Table S2*.

## Benchmarking via IBD-based downstream analyses

At a higher level, we benchmarked different IBD callers by comparing downstream estimates based on true IBD sets *versus* inferred IBD sets. These IBD-based estimates include positive selection scans, $N_e$ estimates, and population structure inference via the community detection algorithm InfoMap.

We scanned for positive selection signals using the IBD-based thresholding method followed by validation with integrated haplotype score-based statistics $X_{iHS}$ as previously described (*Guo et al., 2024*).

We inferred the trajectory of effective population size, $N_e$, for the last 100 generations using `IBDNe`. As this method uses IBD shared by diploid individuals as input, we converted each haploid genome to pseudo-homozygous diploid individuals. We inferred the trajectory $N_e$ using most of the default parameters except for setting the minregion parameter to 10 cM to allow the inclusion of short contigs in the analysis. The final estimates are scaled by 0.25 to compensate for the haploid-to-diploid conversion. By default, we used the value of 2 cM for the mincm parameter to only include IBD segments ≥ 2 cM for inferring the $N_e$ trajectory; as indicated in the Results section, we also set mincm to 4 to test whether excluding short IBD segments can improve the accuracy of $N_e$ estimates. As the `IBDNe` algorithm does not work well when IBD segments shared by close relatives are included in the input, we followed the procedure described in the original work by *Browning and Browning, 2015*. For simulated data, we utilized the TMRCA information of the true IBD segments to filter the IBD segments before calling `IBDNe`. For true IBD segments (generated by `tskibd`), we excluded IBD segments with TMRCA < 1.5; for inferred IBD segments (called by `hap-IBD`, `hmmIBD`, `isoRelate`, `Refined IBD`, `phased IBD`), we removed (inferred) segments that overlap with any true IBD segment with TMRCA < 1.5 shared by the same genome pair. For empirical data where true IBD and TMRCA are not available, we pruned highly related isolates by iteratively removing the genome that has the highest number of close relatives defined by pairwise total IBD > 0.5 of genome size until no close relatives are present in the remaining subgroup (*Guo et al., 2024*).

For population structure inference, we first built the pairwise total IBD matrix, each element being the total IBD for a pair of genomes. The matrix was then squared and used as a weighted adjacency matrix to construct an IBD-sharing network. We then ran the InfoMap algorithm to infer community membership. Genomes assigned the same membership were inferred to be of the same subpopulation. We calculated an adjusted Rand index using the `igraph-python` package (*Csardi and Nepusz, 2006*) to analyze the agreement between true population labels and inferred community labels. For empirical data, we excluded IBD segments shorter than 4 cM when calculating the total IBD matrix to help reduce noise due to false positives and set each element with a value < 5 to zero in the unsquared IBD matrix to decrease the density of the IBD matrix.

Before all high-level benchmarking analyses, we pruned highly related samples as mentioned above. As IBD-based estimates can be biased by strong positive selection in empirical data, we removed high IBD-sharing peaks with peak impact index > 0.01 as previously described (*Guo et al., 2024*) before any downstream analyses.

## Processing empirical data sets

We constructed empirical data sets for validation using genotype data from whole-genome sequencing samples from the MalariaGEN *Pf7* database (*Abdel Hamid et al., 2023*). We used *malariagen_data* 7.13 to download high-quality monoclonal samples that pass quality control. Monoclonal samples were determined by $F_{ws} > 0.95$ ($F_{ws}$ table available from the MalariaGEN website). Quality control labels were extracted directly from the metadata (provided in the *malariagen_data* package).

We then generated haploid genomes (phased genotype data) using the dominant allele from genotype calls. The dominant allele of each genotype call was determined by the per-sample allele depth (AD) fields. For each sample and site, the allele supported by 90% of total reads (total AD values) in a genotype call with at least 5 total reads was used as the dominant allele. Genotype calls without dominant alleles were marked as missing; those with dominant alleles were replaced with a phased genotype homozygous for the dominant allele. The genotype data were further filtered by sample missingness and SNP minor allele frequency and missingness. The resulting genotype data had per-SNP and per-sample missingness < 0.1 and minor allele frequency ≥ 0.01. Genotypes based on dominant alleles were further imputed without a panel using `Beagle` 5.1 (*Browning et al., 2018*). These processing steps generated phased, imputed, pseudo-homozygous diploid genotype data, ready for IBD detection.

We constructed different data sets, including two 'single' population data sets and a 'structured, multiple' population data set, by subsampling the above haploid genomes according to sampling time and location. For each data set, we set the time window to 2-3 years to reduce the sample time heterogeneity and then shifted the window within all possible sampling years and chose one that maximized the sample size. For the 'single' population data sets, we further restricted the sample locations to a relatively small geographic region, such as eastern Southeast Asia, as the data set was used for $N_e$ estimation, which assumes a homogeneous population. For the 'multiple' population data set, we included samples from different continental or subcontinental regions using 'Population' labels from the meta-information table provided with the MalariaGEN *Pf7* database. To make the sample size of each 'population' more balanced, we set a maximum number of samples of 300. Populations with samples larger than 300 are subsampled to a size of 300; populations with a size smaller than 100 were not included in the multiple-population data set. The details of the sampling location and time information were summarized in *Supplementary file 2—Tables S3, S4 and S5*.

Details about the preparation and analysis of the empirical data sets can be found at https://github.com/bguo068/bmibdcaller_empirical (copy archived at *Guo, 2025c*).

## Measuring computational runtime and memory usage

The genomes were simulated with $N_0 = 1000$ and $s = 0.0$ under the single-population model. The runtime and maximum memory usage were measured using the GNU time 1.7 utility. To allow a more appropriate comparison, we ensured: (1) the time used for data pre- and post-processing was excluded; (2) memory resource allocation was capped at 30 gigabytes per IBD call; (3) input genotype data included only common SNPs with minor allele frequency ≥ 0.01; (4) the minimum reported IBD segment length was set to 2.0 cM.

## Replications and uncertainty of measures

Replications of simulations are conducted at two levels: (1) Replication of simulation sets: For each combination of simulation parameters, we performed n = 3 full sets of simulations of populations and sampled n = 1000 haploid genomes per population. (2) Each of the 14 chromosomes in the genomes of a population was simulated independently, which are replicates of each other (see *Supplementary file 1—Data S1* for detailed simulation parameters and replications at simulation set and chromosome levels).

Different analyses reported the uncertainty of measures at various levels of replications or units of observations (as mentioned in the corresponding figure legends and *Supplementary file 2—Data S2*). The choice was made based on the level of uncertainty most relevant to the measures. (1) For IBD accuracy-based IBD segment overlapping analysis, the mean ± standard deviation (SD) was calculated at the segment level for IBD segment false positive and false negative rates for each length bin, or at the genome-pair level for IBD genome-wide error rates. (2) For IBD-based genetic relatedness, the uncertainty is directly visualized in scatter plots at the genome-pair level. (3) For IBD-based selective

signal scans, the mean ± SD of the number of true selection signals (peaks) and false selection signals were calculated at the simulation set level (n = 3 full simulation sets). (4) For IBD network community detection, the mean ± SD of the adjusted Rand index was reported at the simulation set level (n = 3). (5) For IBD-based $N_e$ estimates, bootstrap confidence intervals were obtained directly from `IBDNe` using a single simulation set. (6) For the measure of computational efficiency and memory usage, the mean ± SD was calculated across chromosomes from the same simulation sets.

Given our large sample size of 1000 haploid genomes, the uncertainty reported at the simulation set level is relatively small and can be measured with a limited number of replications. Additionally, full sets of simulation replications were computationally intensive. Therefore, we opted to run n = 3 full simulation sets when it was necessary to measure uncertainty at the simulation set level. For measures for which uncertainty was reported at the segment level or genome-pair levels, only results from a representative simulation set were reported if the results were consistent across n = 3 simulation sets.

## Code availability

Custom tools or scripts were provided in the following GitHub repositories: (1) bmibdcaller_simulations: a Nextflow pipeline to benchmark different IBD detection methods and optimize IBD caller-specific parameters by simulating *Plasmodium falciparum*-like genomes and using true IBD (https://github.com/bguo068/bmibdcaller_simulations, v0.1.0, copy archived at *Guo, 2025a*). (2) bmibdcaller_empirical: a Nextflow pipeline to benchmark IBD callers with empirical data by comparing IBD-based estimates with expected patterns (https://github.com/bguo068/bmibdcaller_empirical, v0.1.0, copy archived at *Guo, 2025c*). (3) ishare/ibdutils: a Rust crate and command-line tools designed to facilitate the analysis of rare-variant sharing and identity-by-descent (IBD) sharing (used here mainly for fast IBD segment overlapping analysis) (https://github.com/bguo068/ishare, copy archived at *Guo, 2025b*, v0.1.11, command-line tool `ibdutils`).

## Acknowledgements

This publication uses MalariaGEN data as described in "Pf7: an open dataset of *Plasmodium falciparum* genome variation in 20,000 worldwide samples" MalariaGEN et al, Wellcome Open Research 2023, 8:22 https://doi.org/10.12688/wellcomeopenres.18681.1. This work was supported by NIH 1R01AI145852 granted to ST-H and TDO by the U.S. National Institutes of Health.

## Additional information

### Funding

| Funder | Grant reference number | Author |
| --- | --- | --- |
| National Institutes of Health | 1R01AI145852 | Shannon Takala-Harrison Timothy D O'Connor |

The funders had no role in study design, data collection and interpretation, or the decision to submit the work for publication.

### Author contributions

Bing Guo, Conceptualization, Software, Formal analysis, Visualization, Writing – original draft; Shannon Takala-Harrison, Timothy D O'Connor, Supervision, Funding acquisition, Writing – review and editing

### Author ORCIDs

Bing Guo https://orcid.org/0000-0002-3998-5981
Shannon Takala-Harrison https://orcid.org/0000-0003-4674-8500
Timothy D O'Connor https://orcid.org/0000-0002-0276-1896

Reviewer #1 (Public review): https://doi.org/10.7554/eLife.101924.3.sa1
Reviewer #2 (Public review): https://doi.org/10.7554/eLife.101924.3.sa2
Author response https://doi.org/10.7554/eLife.101924.3.sa3

## Additional files

### Supplementary files
MDAR checklist

Supplementary file 1. Detailed simulation and IBD calling parameters and values . (**Data S1**) The Excel file contains 18 tabs. Tab 1 provides a summary of the tables, including a schematic of the parameter combinations for 3 demographic models and 5 IBD callers, parameter descriptions, and a list of the names of the other tabs. Tab 2 details the simulation replications at the chromosome and simulation set levels. Tab 3 presents all parameter values used in the simulations and IBD detection. Tabs 4-18 contain the parameter values for each demographic model-IBD caller combination.

Supplementary file 2. Supplementary tables and data. (**Table S1**) The default and optimized values for parameters used in inferring IBD segments via different callers. A link to the source code and a citation of the corresponding article are provided for each IBD caller. For `hap-IBD`, `Refined IBD`, and `hmmIBD`, the parameters are used on the command line except that the parameter rec_rate of `hmmIBD` needs to be specified in the source code file `hmmIBD`.c. For `phased IBD` and `isoRelate`, the parameters are specified within a Python or R script. The details of how the parameters are specified can be found in the scripts on GitHub (https://github.com/bguo068/bmibdcaller_simulations/tree/main/bin). Note that mincm and minmaf are values shared across IBD callers to allow fair comparisons. (**Table S2**) Grid/Line search to optimize IBD caller-specific parameters. Column 2 lists the optimized parameters (other parameters are not explored). Column 3 shows the tested values for each parameter. Some parameters are optimized via two steps, such as coarse search (coarse) and fine-tuning (fine-tune). Column 4 provides comments on the impact of the values on IBD accuracy (measured as false positive rates (FP) and false negative rates (FN)). (**Table S3**) Isolates in the 'multi-population' data set used in empirical validation. Rows are counts of isolates from different locations ('Population' labels from MalariaGEN P*f*7 meta information table). (**Table S4**) Isolates in the 'single-population' data set for 'AS-SE-E' population used in empirical validation. Rows are counts of isolates from different locations ('Country' labels from MalariaGEN P*f*7 meta information table); columns are counts of isolates collected in a given year. (**Table S5**) Isolates in the 'single-population' data set for 'AF-W-Ghana' population used in empirical validation. Rows are counts of isolates from different locations ('admin level 1' labels from MalariaGEN P*f*7 meta information table); columns are counts of isolates collected in a given year. (**Data S2**) Detailed IBD-level benchmarking results shown in heatmaps. Each panel (a-r) represents the FN/FP rates for a specific combination of IBD callers (`hap-IBD`, `hmmIBD`, `isoRelate`, `phased IBD`, and `Refined IBD`), demographic models (single population model, multiple-population model, and UK human demographic model), and recombination rates (human versus P*f*) as indicated in the text above the panel. Note that benchmarking for genomes with human recombination rates was not performed for hmmIBD and isoRelate as they did not scale well for large genome sizes (in base pairs). For each heatmap, the searched parameters and their values are indicated as the $x$ and $y$ labels and tick labels, respectively; the labels in gray background on the top indicate the IBD length bin that was used to calculate FN/FP rates (labels in gray on the left); the bold labels in gray at the bottom show either different groups (e.g. coarse search or fine-tune) or the third parameter searched (such as min-maf=0.01 and min-maf=0.00).

### Data availability
All empirical data used was publicly available from MalariaGEN Pf7 (https://www.malariagen.net/resource/34/; *Abdel Hamid et al., 2023*).

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
