## [Editor Report · eLife Assessment]

This **important** study presents an evaluation of several tools used for detecting Identity-By-Descent (IBD) segments in highly recombining genomes, using simulated data to replicate the high recombination and low marker density of *Plasmodium falciparum*, the parasite responsible for malaria. The evidence presented by the authors is **convincing** demonstrating that users should be cautious calling IBD when SNP density is low and recombination rate is high. This study will be of interest to scientists working in the field of genome evolution and infectious diseases

---

## [Referee Report · Reviewer #1 (Public review)]

Summary:

Authors benchmarked five IBD detection methods (hmmIBD, isoRelate, hap-IBD, phasedIBD, and Refined IBD) in *Plasmodium falciparum* using simulated and empirical data. *Plasmodium falciparum* has a mutation rate similar to that of humans but a much higher recombination rate and lower SNP density. Thus, the authors evaluated how recombination rate and marker density affect IBD segment detection. Next, they performed parameter optimization for *Plasmodium falciparum* and benchmarked the robustness of downstream analyses (selection detection and Ne inference) using IBD segments detected by each method. They also tracked the computational efficiency of these methods. The authors' work is valuable for the tested species, and the analyses presented support their claim that users should be cautious when calling IBD in contexts of low SNP density and high recombination rate.

Strengths:

The study design is convincing and well-structured. The authors chose to use *P. falciparum*, which presents an interesting case due to its high recombination rate and a mutation rate similar to that of humans. The authors note that despite the widespread use of IBD for genomic surveillance, comprehensive evaluation of these methods in high-recombination, low-marker-density contexts has been lacking. Furthermore, they also examined the performance of IBD detection methods developed specifically for *P. falciparum*, and evaluated it with phased data which broadened the applicability of the work.

Weaknesses:

The authors thoughtfully addressed our prior concerns by (1) expanding the simulations; (2) updating figures and methods for clarity; and (3) more clearly framing the broader utility of their benchmarking effort. These updates strengthen the manuscript and make the relevance of their findings beyond *Plasmodium falciparum* more apparent.

More specifically:

The authors added three full replicates per simulation scenario and updated figures to reflect uncertainty at relevant levels, which addresses earlier concerns about reproducibility. The limited number of replicates is due to computational intensity. In the future, broader generalizability and deeper exploration of variance in benchmarking accuracy across parameter space would further strengthen the conclusions/generalizability. The author's also emphasized that, while the study is centered on *Plasmodium falciparum*, the benchmarking framework, not the parameters, are broadly applicable to other sexually recombining species. Lastly, they extensively updated multiple figures to include simulation models, results from simulation replicates, and improved the figures from the previous version of the manuscript.

---

## [Referee Report · Reviewer #2 (Public review)]

Summary:

Guo et al. benchmarked and optimized methods for detecting Identity-By-Descent (IBD) segments in *Plasmodium falciparum* (Pf) genomes, which are characterized by high recombination rates and low marker density. Their goal was to address the limitations of existing IBD detection tools, which were primarily developed for human genomes and do not perform well in the genomic context of highly recombinant genomes. They first analysed various existing IBD callers, such as hmmIBD, isoRelate, hap-IBD, phased-IBD, and refinedIBD. They focused on the impact of recombination on the accuracy, which was calculated based on two metrics, the false negative rate and the false positive rate. The results suggest that high recombination rates significantly reduce marker density, leading to higher false negative rates for short IBD segments. This effect compromises the reliability of IBD-based downstream analyses, such as effective population size (Ne) estimation.

They showed that the best tool for IBD detection in Pf is hmmIBD, because it has relatively low FN/FP error rates and is less biased for relatedness estimates. However, this method is less computationally efficient.

Their suggestion is to optimize human-oriented IBD methods and use hmmIBD only for the estimation of Ne.

Strengths:

Although I am not an expert on *Plasmodium falciparum* genetics, I believe the authors have developed a valuable benchmarking framework tailored to the unique genomic characteristics of this species. Their framework enables a thorough evaluation of various IBD detection tools for non-human data, such as high recombination rates and low marker density, addressing a key gap in the field.

This study provides a comparison of multiple IBD detection methods, including probabilistic approaches (hmmIBD, isoRelate) and IBS-based methods (hap-IBD, Refined IBD, phased IBD). This comprehensive analysis offers researchers valuable guidance on the strengths and limitations of each tool, allowing them to make informed choices based on specific use cases. I think this is important beyond the study of Pf.

The authors highlight how optimized IBD detection can help identify signals of positive selection, infer effective population size (Ne), and uncover population structure.

They demonstrate the critical importance of tailoring analytical tools to suit the unique characteristics of a species. Moreover, the authors provide practical recommendations, such as employing hmmIBD for quality-sensitive analyses and fine-tuning parameters for tools originally designed for non-*P. falciparum* datasets before applying them to malaria research.

Overall, this study represents a meaningful contribution to both computational biology and malaria genomics, with its findings and recommendations likely to have an impact on the field.

Weaknesses:

One weakness of the study is the lack of emphasis on the broader importance of studying *Plasmodium falciparum* as a critical malaria-causing organism. Malaria remains a significant global health challenge, causing hundreds of thousands of deaths annually.

While the study provides a thorough technical evaluation of IBD detection methods and their application to Pf, it does not adequately connect these findings to the broader implications for malaria research and control efforts. Additionally, the discussion on malaria and its global impact could have framed the study in a more accessible and compelling way, making the importance of these technical advances clearer to a broader audience, including researchers and policymakers in the fight against malaria. In the revised version, the authors have placed greater emphasis on this aspect, while still maintaining the methodological focus of the paper.

---

## [Author Response]

The following is the authors’ response to the original reviews

**Public Reviews:**

**Reviewer #1 (Public review):**
Summary:Authors benchmarked 5 IBD detection methods (hmmIBD, isoRelate, hap-IBD, phasedIBD, and Refined IBD) in *Plasmodium falciparum* using simulated and empirical data. *Plasmodium falciparum* has a mutation rate similar to humans but a much higher recombination rate and lower SNP density. Thus, the authors evaluated how recombination rate and marker density affect IBD segment detection. Next, they performed parameter optimization for *Plasmodium falciparum* and benchmarked the robustness of downstream analyses (selection detection and Ne inference) using IBD detected by each of the methods. They also tracked the computational efficiency of these methods. The authors work is valuable for the tested species and the analyses presented appear to support their claim that users should be cautious calling IBD when SNP density is low and recombination rate is high.Strengths:The study design was solid. The authors set up their reasoning for using *P. falciparum* very well. The high recombination rate and similar mutation rate to humans is indeed an interesting case. Further, they chose methods that were developed explicitly for each species. This was a strength of the work, as well as incorporating both simulated and empirical data to support their goal that IBD detection should be benchmarked in *P. falciparum*.Weaknesses:The scope of the optimization and application of results from the work are narrow, in that everything is finetuned for *Plasmodium*. Some of the results were not entirely unexpected for users of any of the tested software that was developed for humans. For example, it is known that Refined IBD is not going to do well with the combination of short IBD segments and low SNP density. Lastly, it appears the authors only did one largescale simulation (there are no reported SDs).

We thank the reviewer for highlighting the strengths and weaknesses of the study.

First, we would like to highlight that: (1) while we use *Plasmodium* as a model to investigate the impact of high recombination and low marker density on IBD detection and downstream analyses, our IBD benchmarking framework and strategies are widely applicable to IBD methods development for many sexually recombining species including both *Plasmodium* and non-*Plasmodium* species. (2) Although some results are not completely unexpected, such as the impact of low marker density on IBD detection, IBD-based methods have been increasingly used in malaria genomic surveillance research without comprehensive benchmarking for malaria parasites despite the high recombination rate. Due to the lack of benchmarking, researchers use a variety of different IBD callers for malaria research including those that are only benchmarked in human genomes, such as refined-ibd. Our work not only confirmed that low marker density (related to high recombination rate) can affect the accuracy of IBD detection, but also demonstrated the importance of proper parameter optimization and tool prioritization for specific downstream analyses in malaria research. We believe our work significantly contributes to the robustness of IBD segment detection and the enhancement of IBDbased malaria genomic surveillance.

Second, we agree that there is a lack of clarity regarding simulation replicates and the uncertainty of reported estimates. We have made the following improvements, including (1) running n = 3 full sets of simulations for each analysis purpose, which is in addition to the large sample sizes and chromosomal-level replications already presented in our initial submission, and (2) updating data and figures to reflect the uncertainty at relevant levels (segment level, genome-pair level or simulation set level).

**Reviewer #2 (Public review):**
Summary:Guo et al. benchmarked and optimized methods for detecting Identity-By-Descent (IBD) segments in *Plasmodium falciparum* (Pf) genomes, which are characterized by high recombination rates and low marker density. Their goal was to address the limitations of existing IBD detection tools, which were primarily developed for human genomes and do not perform well in the genomic context of highly recombinant genomes. They first analysed various existing IBD callers, such as hmmIBD, isoRelate, hap-IBD, phased-IBD, refinedIBD. They focused on the impact of recombination on the accuracy, which was calculated based on two metrics, the false negative rate and the false positive rate. The results suggest that high recombination rates significantly reduce marker density, leading to higher false negative rates for short IBD segments. This effect compromises the reliability of IBD-based downstream analyses, such as effective population size (Ne) estimation. They showed that the best tool for IBD detection in Pf is hmmIBD, because it has relatively low FN/FP error rates and is less biased for relatedness estimates. However, this method is less computationally efficient. Their suggestion is to optimize human-oriented IBD methods and use hmmIBD only for the estimation of Ne.Strengths:Although I am not an expert on *Plasmodium falciparum* genetics, I believe the authors have developed a valuable benchmarking framework tailored to the unique genomic characteristics of this species. Their framework enables a thorough evaluation of various IBD detection tools for non-human data, such as high recombination rates and low marker density, addressing a key gap in the field. This study provides acomparison of multiple IBD detection methods, including probabilistic approaches (hmmIBD, isoRelate) and IBS-based methods (hap-IBD, Refined IBD, phased IBD). This comprehensive analysis offers researchers valuable guidance on the strengths and limitations of each tool, allowing them to make informed choices based on specific use cases. I think this is important beyond the study of Pf. The authors highlight how optimized IBD detection can help identify signals of positive selection, infer effective population size (Ne), and uncover population structure. They demonstrate the critical importance of tailoring analytical tools to suit the unique characteristics of a species. Moreover, the authors provide practical recommendations, such as employing hmmIBD for quality-sensitive analyses and fine-tuning parameters for tools originally designed for non-*P. falciparum* datasets before applying them to malaria research.Overall, this study represents a meaningful contribution to both computational biology and malaria genomics, with its findings and recommendations likely to have an impact on the field.Weaknesses:One weakness of the study is the lack of emphasis on the broader importance of studying *Plasmodium falciparum* as a critical malaria-causing organism. Malaria remains a significant global health challenge, causing hundreds of thousands of deaths annually. The authors could have introduced better the topic, even though I understand this is a methodological paper. While the study provides a thorough technical evaluation of IBD detection methods and their application to Pf, it does not adequately connect these findings to the broader implications for malaria research and control efforts. Additionally, the discussion on malaria and its global impact could have framed the study in a more accessible and compelling way, making the importance of these technical advances clearer to a broader audience, including researchers and policymakers in the fight against malaria.

We thank the reviewer for highlighting the need to better contextualize the work and emphasize its relevance to malaria control and elimination efforts. We have edited the introduction and discussion sections to highlight the importance of studying *Plasmodium* as malaria-causing organisms and why IBD-based analysis is important to malaria researchers and policymakers. We believe the changes will better emphasize the public health relevance of the work and improve clarity for a general audience.

We would like to clarify that we are not recommending that researchers “optimize human-oriented IBD methods and use hmmIBD only for the estimation of Ne.” We recommended hmmIBD for Ne analysis; however, hmmIBD can be utilized for other applications, including population structure and selection detection. Thus, we generally recommend using hmmIBD for *Plasmodium* when phased genotypes are available. To avoid potential misunderstandings, we have revised relevant sentences in the abstract, introduction, and discussion. One reason to consider human-oriented IBD detection methods in *Plasmodium* research is that hmmIBD currently has limitations in handling large genomic datasets. Our ongoing research focuses on improving hmmIBD to reduce its computational runtime, making it scalable for large *Plasmodium* wholegenome sequence datasets.

**Recommendations for the authors**:
**Reviewer #1:**
(1) Additional experiments(i) More simulation replicates would be valuable here. The way that results are presented, it appears as though there are no replicates. Apologies if I am incorrect, but when looking through the authors code the --num_reps defaults to one simulation and there are no SDs reported for any figure. Perhaps the authors are bypass replicates by taking a random sample of lineages? Some clarification here would be great.

We agree with the reviewer’s constructive suggestions. We have increased the number of simulation sets to (n = 3) in addition to the existing replicates at the chromosomal level. We did not use a larger n for full sets of simulation replicates for two reasons: (1) full replication is quite computationally intensive (n=3 simulation sets already require a week to run on our computer cluster with hundreds of CPU cores). (2), the results from different simulation sets are highly consistent with each other, likely due to our large sample size (n = 1000 haploid genomes for each parameter combination). The consistency across simulation sets can be exemplified by the following figures (Author response image 1 and 2) based on simulation sets different from Figures and Supplementary Figures included in the manuscript.

**Author response image 1. sa3fig1:** Additional simulation sets repeating experiments shown in Fig 2.

**Author response image 2. sa3fig2:** Post-optimization Ne estimates based on three independent simulation sets (Fig 5 shows data simulation set 1).

In our updated figures, we address the uncertainty of measurements as follows:

(1) For IBD accuracy based on overlapping IBD segments, we present the mean ± standard deviation (SD) at the segment level (IBD segment false positives and false negatives for each length bin) or genome-pair level (IBD error rates at the genome-wide level). Figures in the revised manuscript show results from one of the three simulation set replicates. The SD of IBD segment accuracy is included in all relevant figures. In the S2 Data file, we chose not to show SDs to avoid text overcrowding in the heatmaps; however, a detailed version, including SD plotting on the heatmap and across three simulation set replicates, is available on our GitHub repository at https://github.com/bguo068/bmibdcaller_simulations/tree/main/simulations/ext_data.

(2) For IBD-based genetic relatedness, the uncertainty is depicted in scatterplots.

(3) For IBD-based selection signal scans, we provide the mean ± SD of the number of true selection signals and false selection signals. The SD is calculated at the simulation set level (n=3).

(4) For IBD network community detection, the mean ± SD of the adjusted Rand index is reported at the simulation set level (n=3). A representative simulation set is randomly chosen for visualization purposes.

(5) For IBD-based Ne estimates, each simulation set provides confidence intervals via bootstrapping. We found Ne estimates across n=3 simulation sets to be highly consistent and decided to display Ne from one of the simulation sets.

(6) For the measurement of computational efficiency and memory usage, the mean ± SD was calculated across chromosomes from the same simulation sets.

We have included a paragraph titled "Replications and Uncertainty of Measures" in the methods section to clarify simulation replications. Additionally, a table of simulation replicates is provided in the new S1 Data file under the sheet named “02_simulation_replicates.”

(ii) I might also recommend a table or illustrative figure with all the simulation parameters for the readers rather than them having to go to and through a previous paper to get a sense of the tested parameters.

We have now generated tables containing full lists of simulation/IBD calling parameters. We have organized the tables into two sections: simulation parameters and IBD calling parameters. For the simulations, we are using three demographic models: the single-population (SP) model, the multiple-population (MP) model, and the human population demography in the UK (UK) model, each with different sets of parameters. Parameters and their values are listed separately for each demographic model (SP, MP and UK). For the IBD calling, we have five different IBD callers, each with different parameters. We have provided lists of the parameters and their values separately for each caller. In total, there are 15 different combinations of 3 demographic models in simulation and five callers in IBD detection (Author response image 3). We provide a table for each of the 15 combinations. We also provide a single large table by concatenating all 15 tables. In the combined table, demographic model-specific or IBD caller-specific parameters are displayed in their own columns, with NA values (empty cells) appearing in rows where these parameters are not applied (see S2 Data file).

**Author response image 3. sa3fig3:** Schematic of combined parameters from simulations and IBD detection (also included in the S2 Data file).

(2) Recommendations for improving the writing and presentationOverall, the writing was great, especially the introduction.Three thoughts:(i) It would be great if the authors included a few sentences with guidance on the approach one would take if their organism was not human or *P. falciparum*.

We have updated our discussion with the following statement: “Beyond Plasmodium parasites, there are many other high-recombining organisms such as Apicomplexan species like Theileria, insects like *Apis mellifera* (honeybee), and fungi like *Saccharomyces cerevisiae* (Baker's yeast). For these species, our optimized parameters may not be directly applicable, but the benchmarking framework established in this study can be utilized to prioritize and optimize IBD detection methods in a context-specific manner.”

(ii) I think there was a lot of confusion about the simulations as they were presented between the co-reviewer and I. Clarification on whether there were replicates and how sampling of lineages occurred would be helpful for a reader.

We have added a paragraph with heading “Replications and uncertainty of measures” under the method section to clarify simulation replicates. Please also refer to our response above for more details (Reviewer #1 (1) Additional experiments).

(iii) Maybe we missed it, but could the authors add a sentence or two about why isoRelate performed so poorly (e.g. lines 206-207) considering it was developed for *Plasmodium*? This result seems important.

IsoRelate assumes non-phased genotypes as input; therefore, even if phased genotypes are provided, the HMM model used in isoRelate (distinct from the hmmIBD model) may not utilize them. Below, we present examples of IBD segments between true sets and inferred sets from both isoRelate and hmmIBD, where many small IBD segments identified by tskibd (ground truth) and hmmIBD (inferred) are not detected by isoRelate (inferred), although isoRelate still captures very long IBD segments. These patterns are also illustrated in Fig. 3 and S3 Fig. We acknowledge that isoRelate may outperform other methods in the context of unphased genotypes. However, we chose not to benchmark IBD calling methods using unphased genotypes in simulations, as the results may be significantly influenced by the quality of genotype phasing for all other IBD detection methods. The characterization of deconvolution methods is beyond the scope of this paper. We have added a paragraph in the discussion to reflect the above explanation.

**Author response image 4. sa3fig4:** Example IBD segments inferred by *isoRelate* and *hmmIBD* compared to true IBD segments calculated by *tskibd*.

(3) Minor corrections to the text and figuresLines 105-110 feel like introduction because the authors are defining IBD and goals of work

We have shortened these sentences and retained only relevant information for transition purposes.

Line 121-122 The definition of false positive is incorrect, it appears to be the exact text from false negative

We apologize for the typo and have corrected the definition, so that it is consistent with that in the methods section.

Lines 177-180 feels more like discussion than results

We have removed this sentence for brevity.

Figure 1:Remove plot titles from the figureWrite out number in aThe legend in b overlaps the data so moving that inset to the right would be helpful

We have removed the titles from Figure 1. In Figure 1a, we have changed the format of the y-axis tick labels from scientific notation to integers. In Figure 1b, we have adjusted the size and location of the legend so that it does not overlap with the data points.

Figure 2-3 & S4-5:It was hard to tell the difference between [3-4) and [10-18) because the colors and shapes are similar. It might be worth using a different color or shape for one of them?

We have changed the color for the [10-18) group so that the two groups are easier to distinguish.

Figure 3 & S3-5:Biggest suggestion is that when an axis is logged it should not only be mentioned in the caption but also should be shown in the figure as well.

We have updated all relevant figures so that the log scale is noted in the figure captions (legends) as well as in the figures (in the x and/or y axis labels).

Supplementary Figure S2(i) It would be nice to either combine it with the main text Figure 1 (I don't believe it would be overwhelming) or add in the other two methods for comparison

We have now plotted data for all five IBD callers in S1 Fig for better comparison.

(ii) the legend overlaps the data so relocating it to the top or bottom would be helpful

We have moved the legend to the bottom of the figure to avoid overlap with the data.

**Reviewer #2:**
I don't have any major comments on the paper. It is well-written, although perhaps a bit long and repetitive in some sections. Make sure not to repeat the same concepts too many times.

We have consolidated and removed several paragraphs to reduce repetition of the same concepts.

I am not a methodological developer, but it seems you have addressed several challenges regarding IBD detection in *P. falciparum*. You have also acknowledged the study's caveats, which I agree with.

Thank you for the positive comments.

Minor comments:-In my opinion, the paper would benefit from including the workflow figure in the main text rather than keeping it in the supplementary materials. This would make it more accessible and useful for readers.

We have moved the original S1 Fig to be Fig 1 in the main text.

-Some of the figures (e.g. Fig. 2, 4) should be larger for better clarity and interpretation.

We have updated Fig 2 and Fig 4 (now labeled as Figure 3 and 5) to make them larger for improved clarity and interpretation.

-While the focus on *P. falciparum* is understandable, it would have been valuable to include examples of other species and discuss the broader implications of the findings for a broader field.

We have updated the third-to-last paragraph to discuss implications for other species, such as Apicomplexan species like *Theileria*, insects like *Apis mellifera* (honeybee), and fungi like *Saccharomyces cerevisiae* (Baker's Yeast). We acknowledge that optimal parameters and tool choices may vary among species due to differences in demographic history and evolutionary parameters. However, we emphasize that the methods outlined are adaptable for prioritizing and optimizing IBD detection methods in a context-specific manner across different species.

-Figure 6 is somewhat confusing and could use clearer labeling or additional explanation to improve comprehension.

We have updated the labels and titles in the figure to improve clarity. We also edited the figure caption for better clarity.

-Although hmmIBD outperformed other tools in accuracy, its computational inefficiency due to single-threaded execution poses a significant challenge for scaling to large datasets. The trade-off between accuracy and computational cost could be discussed in more detail.

We have added a paragraph in the discussion section to highlight the trade-off between accuracy and computation cost. We noted that we are developing an adapted tool to enhance the hmmIBD model and significantly reduce the runtime via parallelizing the IBD inference process.